# Detection of *Anaplasma* and *Ehrlichia* bacteria in humans, wildlife, and ticks in the Amazon rainforest

Marie Buysse [1,9], Rachid Koual[1,9], Florian Binetruy[1], Benoit de Thoisy[2,3], Xavier Baudrimont[4], Stéphane Garnier[5], Maylis Douine[6], Christine Chevillon [1], Frédéric Delsuc [7], François Catzeflis[7], Didier Bouchon [8] & Olivier Duron [1] ✉

Tick-borne bacteria of the genera *Ehrlichia* and *Anaplasma* cause several emerging human infectious diseases worldwide. In this study, we conduct an extensive survey for *Ehrlichia* and *Anaplasma* infections in the rainforests of the Amazon biome of French Guiana. Through molecular genetics and metagenomics reconstruction, we observe a high indigenous biodiversity of infections circulating among humans, wildlife, and ticks inhabiting these ecosystems. Molecular typing identifies these infections as highly endemic, with a majority of new strains and putative species specific to French Guiana. They are detected in unusual rainforest wild animals, suggesting they have distinctive sylvatic transmission cycles. They also present potential health hazards, as revealed by the detection of *Candidatus* Anaplasma sparouinense in human red blood cells and that of a new close relative of the human pathogen *Ehrlichia ewingii*, *Candidatus* Ehrlichia cajennense, in the tick species that most frequently bite humans in South America. The genome assembly of three new putative species obtained from human, sloth, and tick metagenomes further reveals the presence of major homologs of *Ehrlichia* and *Anaplasma* virulence factors. These observations converge to classify health hazards associated with *Ehrlichia* and *Anaplasma* infections in the Amazon biome as distinct from those in the Northern Hemisphere.

Ehrlichiosis and anaplasmosis are among the tick-borne diseases most frequently reported in humans after Lyme disease[1]. These acute febrile diseases are caused by zoonotic bacteria of the *Ehrlichia* and *Anaplasma* genera that infect mammalian blood cells[2–5]. Human Monocytic Ehrlichiosis (HME) is caused by *E. chaffeensis* that infects monocytes in the peripheral blood, while Human Ewingii Ehrlichiosis (HEE) is caused by its close genetic relative, *E. ewingii*, that rather

infects neutrophils. Human Granulocytic Anaplasmosis (HGA) is caused by *A. phagocytophilum* that also infects neutrophils. The first human cases of ehrlichiosis and anaplasmosis were only reported in 1987 and 1994 respectively, both in North America, but they are now reported worldwide[2,5]. Infections normally cycle between animal reservoirs and ticks, but can be spread to humans by tick bites[2–4]. The clinical manifestations of HME, HEE and HGA are largely non-specific,

[1]MIVEGEC, University of Montpellier, CNRS, IRD, Montpellier, France. [2]Laboratoire des Interactions Virus-Hôtes, Institut Pasteur de Guyane, Cayenne, France. [3]Association Kwata 'Study and Conservation of Guianan Wildlife', Cayenne, France. [4]Direction Générale des Territoires et de la Mer (DGTM) – Direction de l'environnement, de l'agriculture, de l'alimentation et de la forêt (DEAAF), Cayenne, France. [5]Biogéosciences, UMR 6282 uB/CNRS/EPHE, Université Bourgogne Franche-Comté, Dijon, France. [6]Centre d'Investigation Clinique Antilles-Guyane, INSERM 1424, Centre Hospitalier de Cayenne, Cayenne, France. [7]Institut des Sciences de l'Evolution de Montpellier (ISEM), CNRS, IRD, EPHE, Université de Montpellier, Montpellier, France. [8]EBI, University of Poitiers, UMR CNRS 7267, Poitiers, France. [9]These authors contributed equally: Marie Buysse, Rachid Koual. ✉e-mail: olivier.duron@cnrs.fr

including undifferentiated fever with thrombocytopenia, leukopenia, and ranging from subclinical to life-threatening symptoms associated with multi-organ failure[2,3,5]. Ehrlichiosis and anaplasmosis are also the most common tick-borne diseases of domestic animals[6–9]. Bovine anaplasmosis causes high morbidity and mortality in cattle herds, with annual economic losses estimated to be over $300 million in the United States and $800 million in Latin America[10].

The diversity of *Ehrlichia* and *Anaplasma* bacteria has been investigated mainly in humans and domestic animals[6,8,9]. This led to the identification of *E. chaffeensis* as the major *Ehrlichia* species infecting humans, but *E. ewingii* and *E. muris eauclairensis* have also been associated with human ehrlichiosis in North America[3,11,12]. *Anaplasma phagocytophilum* is the major *Anaplasma* species infecting humans[4,13] while another species, provisionally named *A. capra*, is now recognized as an emerging agent of human anaplasmosis in China[14]. Most of the other *Ehrlichia* and *Anaplasma* species are major agents of veterinary diseases as exemplified by *E. ruminantium* (causing heartwater, or cowdriosis), *A. marginale* (bovine anaplasmosis), which together are the most common and lethal tick-borne pathogens in cattle, and *E. canis* (canine ehrlichiosis), which is common in dogs in tropical and subtropical regions[6–9]. Spillover events of some veterinary species (as *A. platys*, *A. bovis*, *A. ovis* and *E. ruminantium*) to humans can occur[15–19] and result in the death of infected patients[15].

The question of how many species of *Ehrlichia* and *Anaplasma* infect humans remains open. Over the last decade, *Ehrlichia* and *Anaplasma* species with links to wildlife and human diseases have continued to be discovered[11,20]. In 2022, a chronic infection by an unknown intraerythrocytic *Anaplasma* species was diagnosed in a splenectomized patient living in the rainforest of French Guiana, South America[20]. The causative agent was a putative species, *Candidatus* Anaplasma sparouinense, distinct to all known *Anaplasma* species but phylogenetically more related to *Candidatus* Anaplasma amazonensis infecting Brazilian sloths[20]. Recent wildlife surveys also revealed the presence of additional putative species and unclassified genovariants with undetermined zoonotic potential[9], especially in South America[21,22]. However, the importance of wildlife in maintaining and spreading emerging infections remains largely unknown with the exception of a few species[23].

The genomes of *Ehrlichia* and *Anaplasma* species share important similarities. As phylogenetically related genera of the Anaplasmataceae family (Rickettsiales), they share a similar global genome architecture and gene content[7,24–28]. As obligate intracellular bacteria, their genomes are reduced (1.2 to 1.5 Mb) with a low metabolic potential[7,24–28]. As intracellular pathogenic bacteria, they encode virulence factors needed to infect host cells, including outer membrane proteins, ankyrin repeat proteins, a type IV secretion system (T4SS) and effector proteins[7,24–27]. However, genomic studies focus mainly on *Ehrlichia* and *Anaplasma* species of major medical and veterinary interest, and current genomic knowledge of infections circulating in wildlife is scarce[7,24–27]. Notably, most descriptions of infections associated with wildlife only use a short gene sequence, while more definitive characterizations of their taxonomy, their phylogenetic positioning and their potential pathogenicity require further genetic and genome sequencing[9,29].

In this study, we report potential health hazards associated with *Ehrlichia* and *Anaplasma* infections in the Amazon biome of French Guiana. To this aim, we conducted a large-scale survey of infections in blood samples of humans living in the depths of French Guianese rainforests, as well as biological samples from wildlife, including wild mammals, birds, and ticks. Using high-throughput bacterial 16S rDNA barcoding, molecular typing, and phylogenetics, we characterized prevalence and genetic diversity of novel genovariants and putative *Ehrlichia* and *Anaplasma* species. Using metagenomics, we further obtained the first genome sequences of *Ca.* Anaplasma sparouinense infecting humans, *Ca.* Anaplasma amazonensis infecting sloths, and a novel putative species, *Ca.* Ehrlichia cajennense, detected in the tick species most commonly biting humans in South America, the Cayenne tick, *Amblyomma cajennense*. We use this genomic data to conduct pangenomic, phylogenomic, and metabolic analyses, with comparisons to major *Ehrlichia* and *Anaplasma* pathogens.

## Results

### Survey of infections

This investigation focuses on French Guiana, South America (Fig. 1), where *Ca.* Anaplasma sparouinense was recently identified as a human tick-borne pathogen[20]. French Guiana is 97% covered by old-growth rainforests and harbors a high level of biodiversity[30,31]. Here, we examined archived individual DNA templates extracted from 1,919 specimens belonging to 72 species sampled in 16 of the 22 communes (administrative divisions) of French Guiana (Fig. 1). This collection includes samples from humans living in gold mining camps located in remote forest ($n = 363$ individuals), 44 species of wild mammals ($n = 626$), five species of passerines ($n = 247$), and 22 species of ticks ($n = 683$) (Fig. 2).

The bacterial 16S rDNA barcoding led to the identification of 18 species infected by *Ehrlichia*, *Anaplasma*, or *Ca.* Allocryptoplasma (that is a sister genus of *Anaplasma*[32]) in 11 of 16 sampled communes (Fig. 1). Infected species include humans, six species of wild mammals, five species of passerines, and six species of ticks (Fig. 2, Supplementary Fig. 1). *Ehrlichia* is detected in 14 species (including four species of wild mammals, five of passerines and five of ticks), *Anaplasma* in eight species (human, three species of wild mammals, and four species of ticks), and *Ca.* Allocryptoplasma in one tick species. Of the 1,919 specimens, we detect 189 positive samples (9.9%) including 64 samples infected by *Ehrlichia* spp. (3.3%), 124 by *Anaplasma* spp. (6.5%), and one by *Ca.* Allocryptoplasma sp. (0.1%) (Fig. 2, Supplementary Fig. 1).

Prevalence of infection is not homogeneous between the 18 infected species (Fig. 2, Supplementary Fig. 1, Source Data). *Ehrlichia* spp. are more prevalent in passerines than in wild mammals as the common opossum, *Didelphis marsupialis*, and the nine-banded armadillo, *Dasypus novemcinctus* (Fisher's exact tests, all $p < 10^{-4}$). *Anaplasma* spp. are more prevalent in brown-throated three-toed sloths, *Bradypus variegatus*, and Linnaeus's two-toed sloths, *Choloepus didactylus*, than in other animals (all $p < 10^{-6}$). In the *B. variegatus* sloth, *Anaplasma* infections are consistently detected at prevalence higher than 50% in ancient (1994-1995) and more recent (2016) blood samples (Source Data). For the blood samples obtained from 363 humans, a single *Anaplasma* infection is detected in one sample who was from the same patient that led to the primary description of *Ca.* Anaplasma sparouinense[20]. Examination of unpublished thin blood films from this patient confirms that *Ca.* Anaplasma sparouinense is an exclusive intraerythrocytic pathogen with 0.3–0.4 μm inclusions located at the periphery of one third of red blood cells (Supplementary Fig. 3A-C), corroborating early observations of other thin blood films[20]. Neither *Ehrlichia* nor *Anaplasma* is detected in blood samples of the 362 other humans (Fig. 2, Supplementary Fig. 1, Source Data).

### Genetic diversity of infections

Sequencing of a large bacterial 16S rDNA fragment (1187–1202 bp) from the 189 infected samples reveals the presence of 16 genovariants, including 10 genovariants of *Ehrlichia* spp. (96.6-99.6% pairwise nucleotide identities), five genovariants of *Anaplasma* spp. (97.5-99.9% pairwise nucleotide identities), and one genovariant of *Ca.* Allocryptoplasma sp. (92.1–93.1% and 94.8–95.4% pairwise nucleotide identities with *Ehrlichia* and *Anaplasma* genovariants, respectively) (Supplementary Table 1). Few of these genovariants are very similar, such as *Ehrlichia* genovariants #9 and #10 (99.65% nucleotide identity) but as their nucleotide sequences differ at four diagnostic positions, they are then considered non-identical.

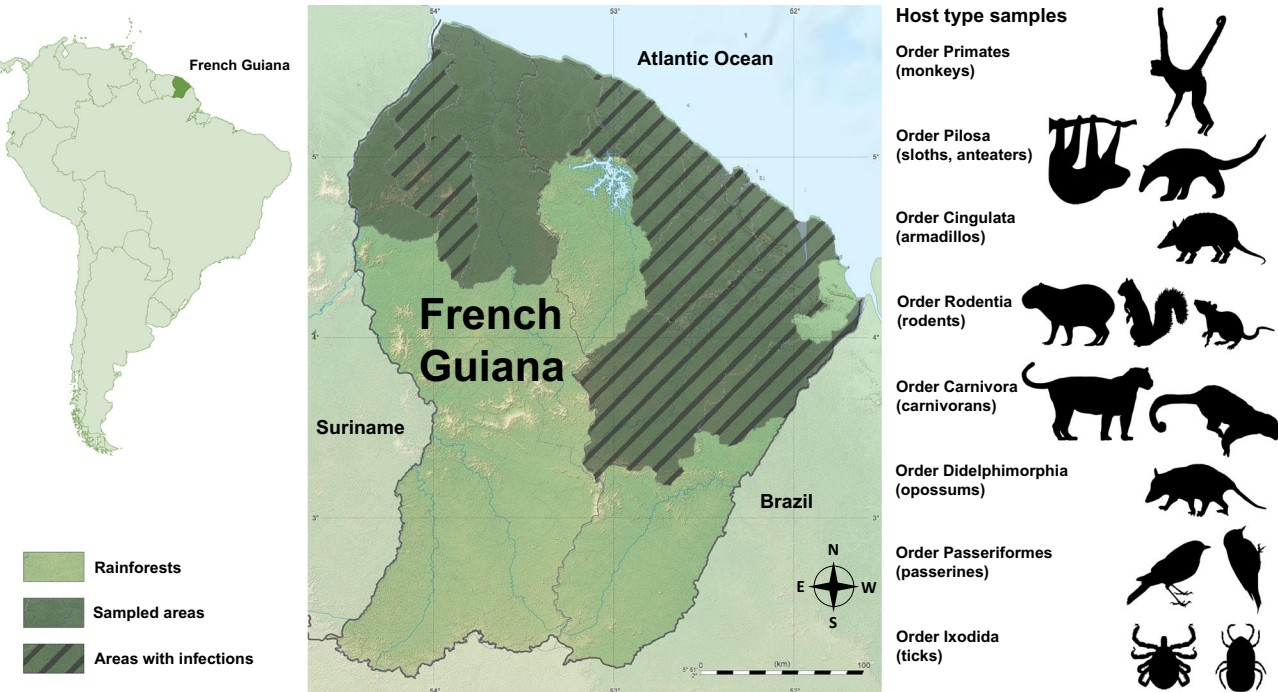

**Fig. 1 | Sampling locations of wild mammals, birds, and ticks in French Guiana.** Grey areas indicate the 16 communes (administrative divisions) where samples were collected. Hatched areas indicate the 11 communes where infections with *Ehrlichia*, *Anaplasma*, and *Ca.* Allocryptoplasma spp. were detected. Source data on infection distribution are provided in the Source Data. The French Guiana base map is available under the license CC BY-SA 3.0 (https://creativecommons.org/licenses/by-sa/3.0) by Eric Gaba via Wikimedia Commons (https://commons.wikimedia.org/wiki/File:Guyane_department_relief_location_map.jpg).

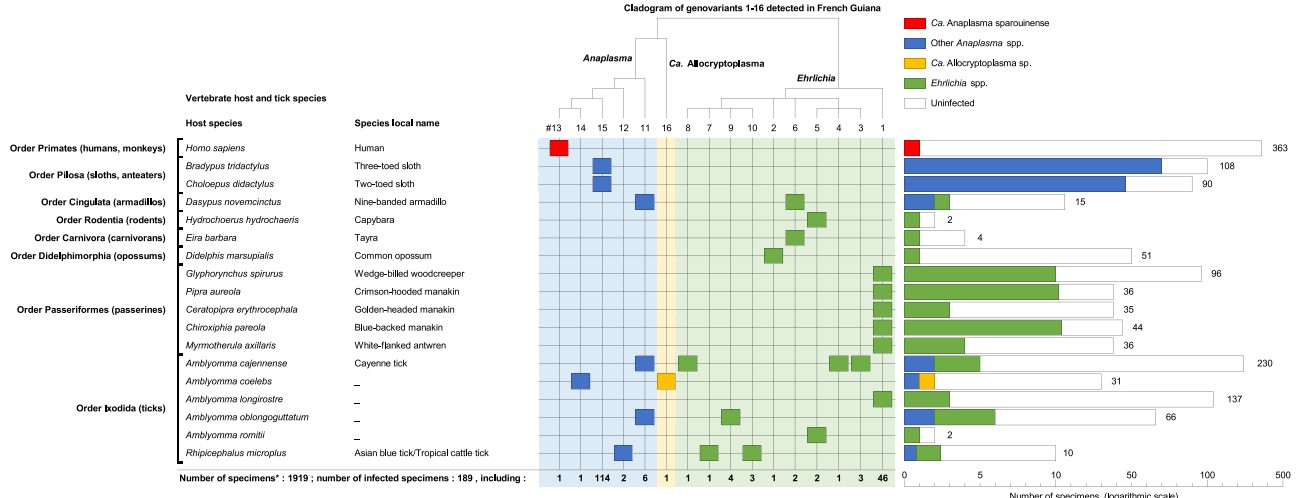

**Fig. 2 | Distribution of *Ehrlichia*, *Anaplasma*, and *Ca.* Allocryptoplasma infections in humans, wildlife, and associated ticks in French Guiana.** Only the 18 infected species are shown here. A complete figure including both the 18 infected and the 54 uninfected species is provided in Supplementary Fig. 1. The left part of the Figure shows the list of the 18 infected species. The top part shows a cladogram of the *Ehrlichia*, *Anaplasma*, and *Ca.* Allocryptoplasma genovariants detected in this study. The right part of the figure gives the distribution of infections in specimens for each of the 18 infected species (color: infected specimens; white: uninfected specimens). Source data on infection prevalence are provided in the Source Data.

The 16S rDNA sequence of *Ca.* Anaplasma sparouinense (genovariant #13) shows high match with *A. marginale* (97.9%; Supplementary Table 1) across nearly the entire 16S rRNA sequence. However, when comparing it with shorter 16S rRNA sequences available on GenBank, it becomes apparent that *Ca.* Anaplasma sparouinense shares the highest nucleotide identities with genovariants infecting wildlife in Brazil, including an *Anaplasma* genovariant detected in *Amblyomma coelebs* ticks collected on South American coatis, *Nasua nasua* (99.8% pairwise nucleotide identities; GenBank accession no.

MT019560), another *Anaplasma* genovariant of black rats, *Rattus rattus* (99.8%; KY391803), and *Ca.* Anaplasma amazonensis of *B. variegatus* sloths (98.6%; MT199810), *C. didactylus* sloths (98.6%; MT199833), and domestic cats (*Felis catus*; 98.6%; OM069304). All other *Anaplasma* genovariants and species show identities <98% with *Ca.* Anaplasma sparouinense (Supplementary Table 1).

Wild mammals and passerines host four genovariants of *Ehrlichia* and two of *Anaplasma* (Supplementary Table 1). The two species of sloths, *B. variegatus* and *C. didactylus*, harbor *Anaplasma* genovariant

#15, which shows high match with *A. ovis* (97.74%; Supplementary Table 1) across nearly the entire 16S rRNA sequence. When comparing it with shorter 16S rRNA sequences available on GenBank, *Anaplasma* genovariant #15 has a 16S rDNA sequence 100% identical to *Ca*. Anaplasma amazonensis of Brazilian sloth species (MT199828, MT199827), also identified in Brazilian domestic cats (OM069304) and in the South American soft tick *Ornithodoros hasei* (MZ220347). All other *Ehrlichia* and *Anaplasma* genovariants characterized in wild mammals and passerines of French Guiana have 16S rDNA sequences distinct to sequences available in public databases (Supplementary Table 1).

Ticks harbor a higher diversity of infections with eight genovariants of *Ehrlichia*, three of *Anaplasma* and one of *Ca*. Allocryptoplasma (Supplementary Table 1). *Anaplasma* genovariant #12 of the tropical cattle tick, *Rhipicephalus microplus*, is identical to *A. marginale*. *Ehrlichia* genovariant #10 of *Rh. microplus* is also 100% identical to an *Ehrlichia* sp. of Malaysia. *Ehrlichia* genovariant #9 identified in *Rh. microplus* exhibits close similarity to *Ehrlichia* sp. VKAA024, also found in *Rh. microplus*, albeit originating from Malaysia (Supplementary Table 1). Additionally, it shares similarities with a species infecting cattle, *Ehrlichia minasensis* (GenBank OQ136684 and QOHL 01000000). These findings suggest that *Ehrlichia* genovariant #9 may represent a distinct strain of *Ehrlichia minasensis*. *Ehrlichia* genovariant #2 from the common opossum (*Didelphis marsupialis*) exhibits high similarity with the full 16S rRNA gene sequence of *Ehrlichia* sp. MieHl1 (found in *Haemaphysalis longicornis* ticks from Japan; Supplementary Table 1) but further analysis reveals 100% pairwise nucleotide identity with a short 16S rRNA gene fragment obtained from an infected white-eared opossums (*Didelphis albiventris*) in Brazil[33] (GenBank OK605040). Additional typing of *Ehrlichia* genovariant #2 using the *gltA* gene sequence and comparisons with the genovariant of white-eared opossum in Brazil (GenBank OK763036-OK763038) confirm that the same *Ehrlichia* genovariant (100% nucleotide pairwise identity) is circulating in opossums in both French Guiana and Brazil.

All other *Ehrlichia*, *Anaplasma* and *Ca*. Allocryptoplasma genovariants found in ticks are distinct to taxa referenced in public databases for their 16S rDNA sequences. *Ehrlichia* genovariant #6 from the nine-banded armadillo (*Dasypus novemcinctus*) and tayra (*Eira barbara*) are distinct, yet they exhibit high pairwise identity to an *Ehrlichia* genovariant infecting tayra in Brazil (Supplementary Table 1) and to a novel putative species, *Ca*. Ehrlichia dumleri, recently described from ring-tailed coatis (*Nasua nasua*) in Brazil[34] (99.70% pairwise nucleotide identity; GenBank OM530515). Additional *gltA* gene typing of *Ehrlichia* genovariant #6 confirms its genetic similarity to the *Ehrlichia* genovariant infecting tayra in Brazil (99.65% nucleotide pairwise identity; GenBank OM055650) and *Ca*. Ehrlichia dumleri (99.40% nucleotide pairwise identity; GenBank OP819940). This indicates that distinct yet genetically related genovariants of *Ehrlichia* are circulating in populations of small Carnivora in both French Guiana and Brazil. Remarkably, *Anaplasma* genovariant #14 detected in *Am. coelebs* is very similar to *Ca*. Anaplasma sparouinense differing only by two nucleotide substitutions (99.998% nucleotide identity). Two generalist tick species commonly biting humans, *Amblyomma cajennense* and *Amblyomma oblongoguttatum*[35], harbor four genovariants of *Ehrlichia* and one of *Anaplasma* (Supplementary Table 1).

Some of the genovariants detected in vertebrates and ticks are similar, which is indicative of their putative transmission cycles in French Guiana (Supplementary Table 1): (1) *Ehrlichia* genovariant #1 detected in the five passerines species is also detected in a tick species feeding on arboreal vertebrates, *Amblyomma longirostre* (larvae of this species are specialized on passerines[35]), (2) *Ehrlichia* genovariant #5 of capybara, *Hydrochoerus hydrochaeris*, is also found in a tick species feeding exclusively on capybara, *Amblyomma romitii*, (3) *Anaplasma* genovariant #11 of the armadillo *D. novemcinctus* is present in *Am. cajennense* and *Am. oblongoguttatum*. Furthermore, *Ehrlichia*

genovariant #5, identified in both capybara and *Am. romitii*, displays a substantial genetic dissimilarity to *Ca*. Ehrlichia hydrochoerus, a recently proposed species found in Brazilian capybaras[36] (96.11% pairwise nucleotide identities; GenBank MW785879 and MW785880). This diversity pattern suggests that significant differences in the nature of infections can exist among animals of the same vertebrate species originating from distinct regions of South America.

## Phylogenetics

Maximum likelihood (ML) phylogenetic analysis based on the 16S rDNA sequence alignment reveals a robust clustering of the 16 genovariants within the family Anaplasmataceae in the genera *Ehrlichia*, *Anaplasma*, and *Ca*. Allocryptoplasma (Fig. 3, Supplementary Fig. 2). *Ca*. Anaplasma sparouinense, *Anaplasma* genovariant #14 of the tick *Am. coelebs* and *Ca*. Anaplasma amazonensis form a well-supported subclade of *Anaplasma* divergent from all other *Anaplasma* species. Other *Anaplasma* genovariants detected in French Guiana cluster with valid *Anaplasma* species: *Anaplasma* genovariant #12 of the tick *R. microplus* with *A. marginale*, and *Anaplasma* genovariant #11 of the armadillo *D. novemcinctus* and *Amblyomma* spp. Ticks with *A. platys*. *Ca*. Allocryptoplasma genovariant #16 of the tick *Am. coelebs* clusters with *Ca*. Allocryptoplasma genovariants of other tick species from North America, Europe, and Asia. Two *Ehrlichia* genovariants (#7 and #8) of the ticks *Am. cajennense* and *Am. oblongoguttatum* are closely related to the human pathogen *E. ewingii*, while *Ehrlichia* genovariant #10 of the tick *R. microplus* is more related to the cattle pathogen *Ehrlichia minasensis*. However, most *Ehrlichia* genovariants have not clear phylogenetic relationships with other *Ehrlichia* species as best exemplified with *Ehrlichia* genovariant #1 found in passerines and the tick *Am. longirostre*, which forms a subclade distinct from all other species (Fig. 3, Supplementary Fig. 2).

## Metagenomics

We sequenced metagenomes from three infected samples: A human blood sample positive for *Ca*. Anaplasma sparouinense (genovariant #13, strain Sparouine), a blood sample of a brown-throated three-toed sloth *B. variegatus* positive for *Ca*. Anaplasma amazonensis (genovariant #15, strain Petit Saut), and an *Am. cajennense* tick sample positive for an *Ehrlichia* sp. (genovariant #8, *Ca*. Ehrlichia cajennense hereafter, strain Matoury) (Fig. 4, Source Data). On account of distinct genetic traits of genovariant #8 (see below), we propose the designation *Ca*. Ehrlichia cajennense for this novel bacterium. The specific name refers to the Cayenne tick *Am. cajennense*, in which the bacterium has been discovered.

The assemblies of *Ca*. Anaplasma sparouinense and *Ca*. Anaplasma amazonensis Metagenome-Assembled Genomes (hereafter, MAG) (117 and 81 contigs, respectively) are similar in size (1187–1176 Mb), average G + C content (49.43–50.46%), number of predicted protein-coding genes (1039–1051), no prophage or plasmid (Fig. 4A, Supplementary Table 2). As these two MAGs are fragmented and disrupted across several contigs, any absent genes may potentially be located within a genomic gap. However, MAGs of *Ca*. Anaplasma sparouinense and *Ca*. Anaplasma amazonensis have both an estimated completeness of 97.18% and the number of predicted protein-coding genes falls within a similar range to those observed in other *Anaplasma* spp. genomes (Supplementary Table 2). The MAG of *Ca*. Ehrlichia cajennense is a complete circular chromosome with size of 1,177,323 Mb, a 32.08% average G + C content, 1,410 predicted protein-coding genes, no prophage or plasmid (Supplementary Table 2). MAG of *Ca*. Ehrlichia cajennense has an estimated completeness of 90.48% (Supplementary Table 2). However, since this MAG is a circular contig, the score reflects more the substantial variation in gene content among *Ehrlichia* genomes than a misassembled genome. Considering the typical threshold of 96% Average Nucleotide Identity (ANI) uses to

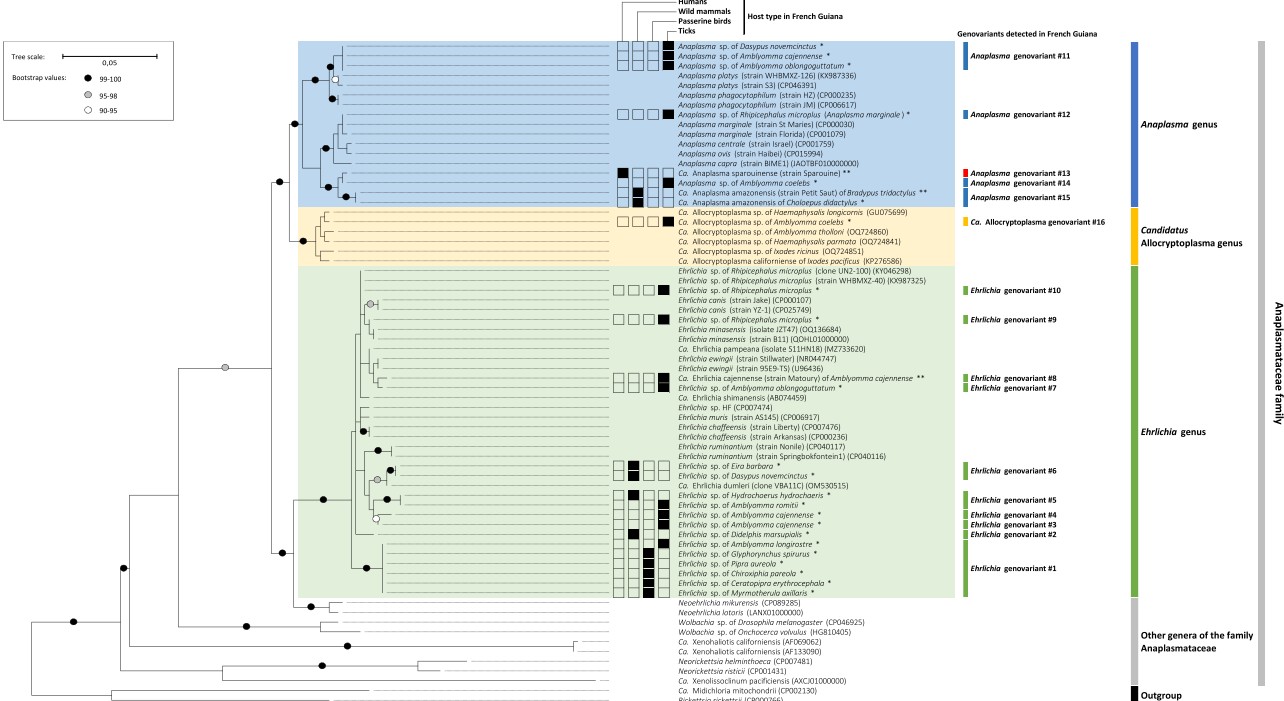

**Fig. 3 | Phylogeny of the family Anaplasmataceae constructed using maximum-likelihood (ML) estimations based on 16S rDNA sequences with a total of 1149 unambiguously aligned bp (best-fit approximation for the evolutionary model: GTR + G + I).** Only one 16S rDNA sequence per genovariant and per host species are shown for data produced in this study. A complete phylogenetic tree displaying all the sequences generated in this study is provided in Supplementary Fig. 2. For *Ehrlichia*, *Anaplasma*, and *Ca.* Allocryptoplasma detected in French Guiana, host type (humans, wild mammals, passerines or ticks) is shown by black squares. *, 16S rDNA sequences of *Ehrlichia*, *Anaplasma*, and *Ca.* Allocryptoplasma obtained in this study; **, *Ca.* Anaplasma sparouinense, *Ca.* Anaplasma amazonensis, and *Ca.* Ehrlichia cajennense MAGs obtained in this study. GenBank accession numbers of other sequences used in analyses are shown on the phylogenetic trees. Numbers at nodes indicate bootstrap support percentage with 1,000 replicates. Only bootstrap values > 90% are shown. The scale bar is in units of mean number of substitutions/site.

define species of Anaplasmataceae[29], *Ca.* Anaplasma sparouinense, *Ca.* Anaplasma amazonensis and *Ca.* Ehrlichia cajennense may each represent a valid and distinct species as further described.

ANI is 88.26% between *Ca.* Anaplasma sparouinense and *Ca.* Anaplasma amazonensis, and both have ANIs lower than 91% with any valid *Anaplasma* species (Supplementary Table 3). Based on orthologous genes, *Ca.* Anaplasma sparouinense and *Ca.* Anaplasma amazonensis share 825 genes while the core genome of the genus *Anaplasma* contains 692 genes (Fig. 4B). Overall, *Ca.* Anaplasma sparouinense and *Ca.* Anaplasma amazonensis exclusively share 34 genes that are not present in other *Anaplasma* species. However, *Ca.* Anaplasma sparouinense contains seven unique genes and *Ca.* Anaplasma amazonensis three unique genes (Fig. 4B). All the genes specific to *Ca.* Anaplasma sparouinense and *Ca.* Anaplasma amazonensis are encoding hypothetical proteins whose function has not been determined. Phylogenomic analysis based on 176 single-copy orthologs (SCOs) confirms that *Ca.* Anaplasma sparouinense and *Ca.* Anaplasma amazonensis cluster together within the genus *Anaplasma* (Fig. 4D). Their closest relatives are the members of the *A. marginale*, *A. centrale*, *A. capra*, and *A. ovis* subclade, while *A. phagocytophilum* and *A. platys* are more distantly related (Fig. 4D).

ANI range from 83.24 to 83.82% between *Ca.* Ehrlichia cajennense and other *Ehrlichia* species (Supplementary Table 4). Based on orthologous genes, *Ca.* Ehrlichia cajennense shares 744 genes with other *Ehrlichia* species but contains 11 unique genes (all encoding for hypothetical proteins) not present in other *Ehrlichia* species (Fig. 4C). *Ca.* Ehrlichia cajennense clusters with other species within the *Ehrlichia* genus with a phylogenetic positioning intermediate between *E. ruminantium* and the subclade formed by *E. canis*, *E. minasensis*, *E. chaffeensis*, and *E. muris* (Fig. 4D). Based on 16S rDNA sequences,

*Ca.* Ehrlichia cajennense is more closely related to *E. ewingii* (Fig. 3) for which no genome is currently available.

## Virulence factors

The most abundant immunogenic outer membrane proteins in *Ca.* Anaplasma sparouinense, *Ca.* Anaplasma amazonensis, and *Ca.* Ehrlichia cajennense are porin proteins from the OMP1 superfamily (usually termed MSP2/P44 in *Anaplasma* spp. and OMP1/P28 in *Ehrlichia* spp.) (Fis. 5A, B). The *Ca.* Anaplasma sparouinense and *Ca.* Anaplasma amazonensis MAGs have nine and 12 *msp2/p44* paralogs, respectively, including eight shared by both MAGs. The *msp2/p44* paralogs are identified on different large contigs, indicating that they are not structured within a large cluster but rather scattered along the *Ca.* Anaplasma sparouinense and *Ca.* Anaplasma amazonensis genomes. Although genome fragmentation hinders the reconstruction of the *msp2/p44* operon structure, it is probable that *msp2/p44* is transcribed from a single operon. This assertion is supported by the presence of single copies of the transcriptional regulator 1 (*tr1*) and three operon-associated genes (*OpAG4*, *OpAG3* and *OpAG2*) (Supplementary Fig. 4). These genes are typically situated downstream of the *msp2/p44* expression locus in some *Anaplasma* genomes[7,24,28]. In *Ca.* Anaplasma sparouinense and *Ca.* Anaplasma amazonensis genomes, an organization of *tr1*, *OpAG4*, *OpAG3* and *OpAG2* loci similar to that observed in *A. marginale* genomes is evident (Supplementary Fig. 4). However, they also exhibit characteristic features of the *A. phagocytophilum* *msp2/p44* operon, such as a nucleoside diphosphate kinase (*ndK*) gene downstream of *tr1* and the association of one *msp2/p44* copy with the outer-membrane protein 1X (*omp1X*) gene. The *msp2/p44* paralogs commonly function as donor sequences, undergoing gene conversion and facilitating the expression of various *msp2/p44* donor sequences at

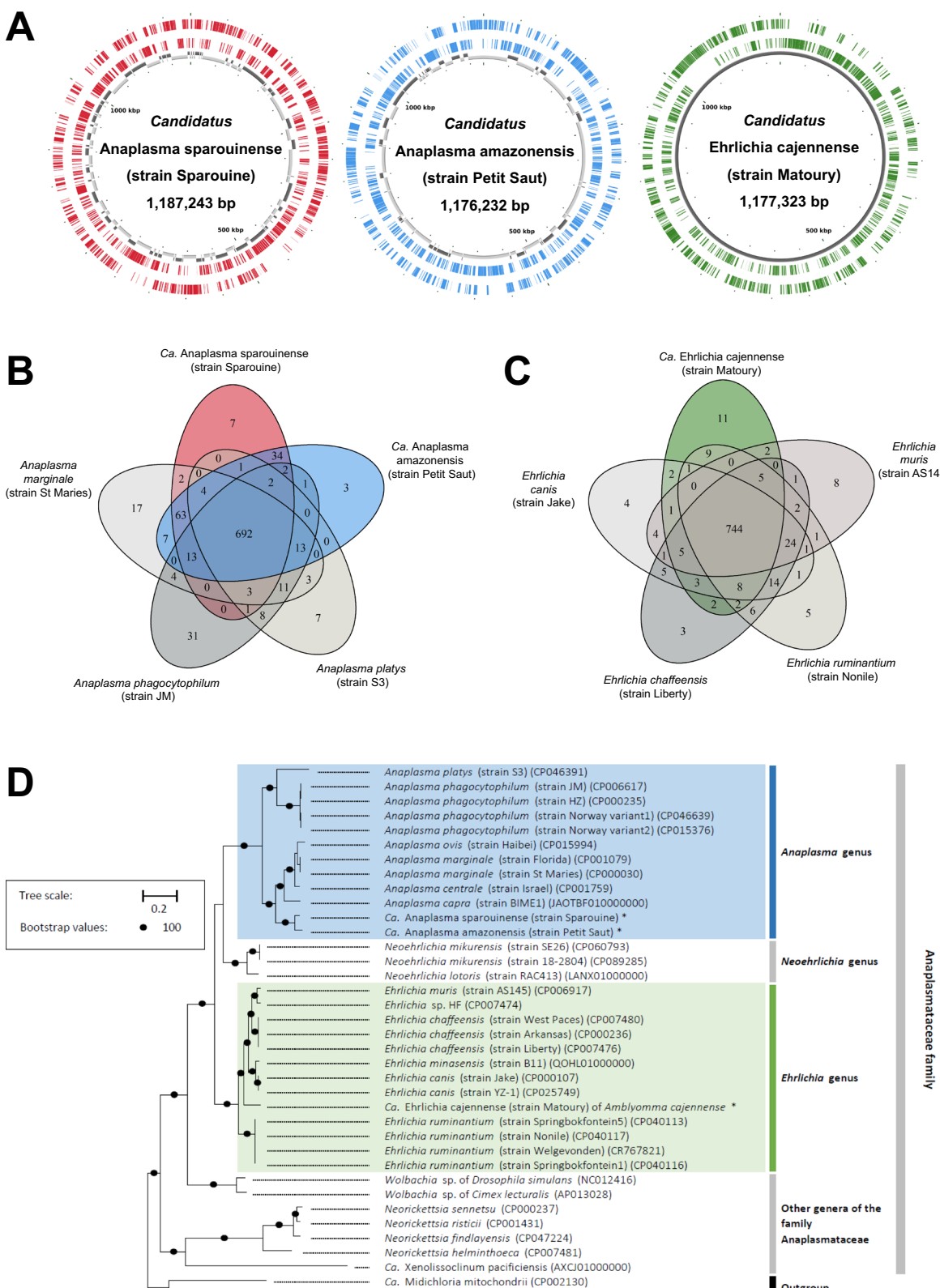

**Fig. 4 | Comparative genomic features of *Ca.* Anaplasma sparouinense, *Ca.* Anaplasma amazonensis, and *Ca.* Ehrlichia cajennense.** **A** Genome maps of *Ca.* Anaplasma sparouinense, *Ca.* Anaplasma amazonensis, and *Ca.* Ehrlichia cajennense. Circles on genome maps correspond to the following (from the edge to the middle): (1) forward strand genes; (2) reverse strand genes; (3) in gray and black, contigs. **B** Venn diagram representing orthologs distribution between *Ca.* Anaplasma sparouinense, *Ca.* Anaplasma amazonensis and representative *Anaplasma* species. **C** Venn diagram representing orthologs distribution among *Ca.* Ehrlichia

cajennense and representative *Ehrlichia* species. **D** Phylogenetic relationships based on whole-genome inferred using maximum likelihood (ML) from a concatenated alignment of 176 single-copy orthologs (37,155 amino acids; best-fit approximation for the evolutionary model: LG + I + G4). The numbers on each node represent the bootstrap support percentage with 1,000 replicates. Only bootstrap values > 90% are shown. The scale bar is in units of mean number of substitutions/site.

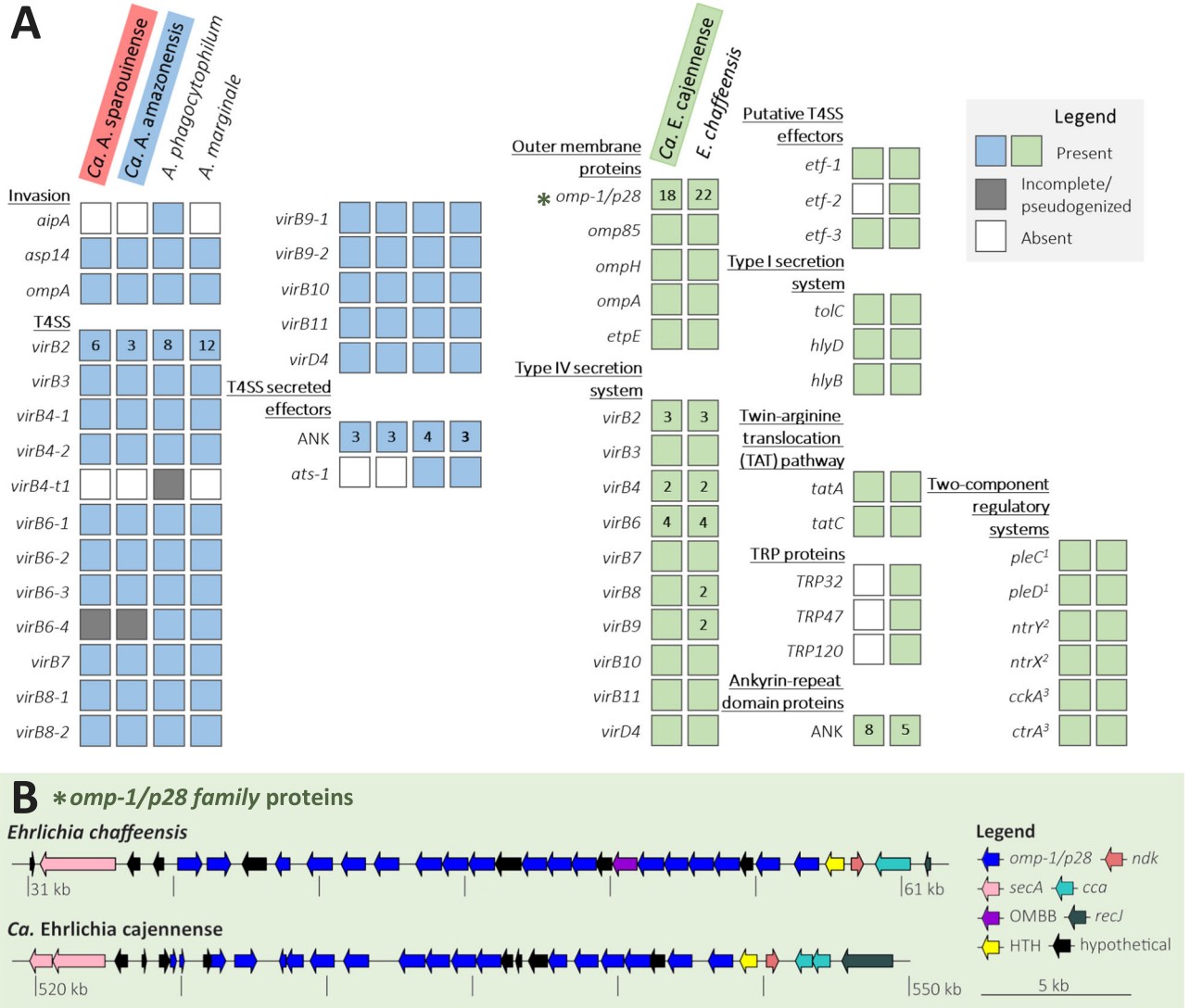

**Fig. 5 | Pathogenic features of *Ca*. Anaplasma sparouinense, *Ca*. Anaplasma amazonensis, *Ca*. Ehrlichia cajennense MAGs and representative genomes of *Anaplasma* and *Ehrlichia* species (*A. phagocytophilum* strain HZ [CP006617], *A. marginale* strain Florida [CP001079], and *E. chaffeensis* strain West Paces [CP007480]). A** Virulence genes in *Anaplasma* spp. and *Ehrlichia* spp. Color of boxes is indicative of the presence or pseudogenization or absence for each gene. Numbers associated with boxes precise the number of gene copies identified, no number is equal to one copy. **B** Structure of the genomic region encoding *omp-1/p28* gene copies in *Ca*. Ehrlichia cajennense MAG, in comparison with a pathogenic representative *E. chaffeensis*. Each intact gene and its direction is represented by an arrow, for which the color is indicative of the protein family (blue: *omp-1/p28*; darkblue: single-stranded-DNA-specific exonuclease RecJ; lightblue: poly A polymerase head domain protein (*cca* gene); red: nucleoside diphosphate kinase Ndk; pink: protein translocase SecA; yellow: helix-turn-helix family protein HTH; violet: outer membrane beta-barrel domain protein OMBB; black: hypothetical protein).

the expression locus, thereby enhancing MSP2/P44 antigenic variation[7,24,28]. The *Ca*. Ehrlichia cajennense MAG has 18 paralogous *omp-1/p28* family located downstream of *tr1*, a putative transcription factor, and upstream of *secA* gene, clustering in a > 29 kb genomic region, similar to the arrangement observed in the *Ehrlichia chaffeensis* genome (Fig. 5B). A homolog of the outer membrane invasin EtpE, used by *E. chaffeensis* to bind mammalian cell receptor DNase X, is also present in *Ca*. Ehrlichia cajennense (Fig. 5A).

The MAGs of *Ca*. Anaplasma sparouinense, *Ca*. Anaplasma amazonensis, and *Ca*. Ehrlichia cajennense each harbors a Sec-independent Type I secretion system (T1SS) and a Type IV secretion system (T4SS), although a few key genes were not detected (Fig. 5A). As the three MAGs are not complete (Supplementary Table 2), the absence of these gene sequences could be due to gaps in the sequencing and therefore to technical artifacts. However, except for *VirB6-4* and *VirB4-t1* in MAGs of *Ca*. Anaplasma sparouinense and *Ca*. Anaplasma amazonensis, and ehrlichial translocated factor 2 (*Etf2*) in *Ca*. Ehrlichia cajennense MAG,

all key genes of the T4SS apparatus are present in the three MAGs, similar to those of other *Anaplasma* and *Ehrlichia* species. The three MAGs also contain effector proteins that bind to mammalian cell machinery and modulate its function, including three to eight domains encoding proteins containing ankyrin (ANK) repeats (*i.e.*, copies of *Ank2* [PFAM12796], *Ank4* [PFAM13637], and ankyrin-like protein PHA03095 and PHA02874). However, no gene encoding tetra-tricopeptide repeat proteins (TPR), used by *E. chaffeensis* to modify mammalian proteins[24], were found in *Ca*. Ehrlichia cajennense MAG[24]. In addition, the *Ca*. Ehrlichia cajennense MAG harbors *Etf1* and *Etf3* genes, which are used by *Ehrlichia* spp. to modulate endosomal and autophagy pathways, reactive oxygen species and apoptosis[24].

## Discussion
This study identifies the tropical rainforests of French Guiana as harboring a high indigenous biodiversity of *Ehrlichia* and *Anaplasma* infections. While most of the observed genovariants are endemic to

this region, they are associated with a wide variety of wild animals and several tick species, including the two species that most frequently bite humans in South America, *Am. cajennense* and *Am. oblongoguttatum*. Furthermore, these infections pose potential pathogenic and zoonotic hazards. Indeed, *Ca.* Anaplasma sparouinense, *Ca.* Anaplasma amazonensis and *Ca.* Ehrlichia cajennense possess virulence factors identified in the *Ehrlichia* and *Anaplasma* species that are pathogenic to humans and domestic animals. The potential of *Ca.* Anaplasma sparouinense to induce intraerythrocytic anaplasmosis in humans is concerning, particularly given that its natural host has yet to be identified. In addition, the detection of a new close genetic relative of the human pathogen *E. ewingii*, *Ca.* Ehrlichia cajennense, in the Cayenne tick reveal potential threats to human health in South America.

Infections with *Ehrlichia* and *Anaplasma* species are potentially ubiquitous in wildlife living in the Amazon biome. In Brazil, already known species and potential new species of *Ehrlichia* and *Anaplasma* have also been detected in wildlife[21,37–40], although most are different from the genovariants detected in this study. At least 16 genovariants are circulating in French Guiana but only two belong to valid species, *A. marginale* and *E. minasensis*, and two to Candidatus species, *Ca.* Anaplasma sparouinense and *Ca.* Anaplasma amazonensis. However, none of the other 14 genovariants we identified in French Guiana had been documented elsewhere. Neither *E. chaffeensis* nor *A. phagocytophilum* were observed, although they are the two most widespread human pathogens of their genera[2–5]. The diversity and endemicity of *Ehrlichia* and *Anaplasma* genovariants in French Guiana, as well as in the northern rainforests of Brazil[21,41], suggest that the health hazard is unique in the Amazon biome. The recent detection of additional genovariants in the Cerrado biome[21,36–40], a vast tropical savanna ecoregion of Brazil, further underscores the distinctiveness of South America in the context of tick-borne diseases. It differs from the health hazard in the Northern Hemisphere where other species are the major etiologic agents of human diseases associated with the genera *Ehrlichia* and *Anaplasma*[3,4,11–13]. Notably, while ruminants are main animal hosts for most *Ehrlichia* and *Anaplasma* species in the Northern Hemisphere[9], some Neotropical *Ehrlichia* and *Anaplasma* genovariants may have evolved specific sylvatic transmission cycles.

In French Guiana, *Ehrlichia* and *Anaplasma* infections are detected in rainforest wildlife animals, including opossums, sloths, armadillos, capybaras, and passerines, and they may depend on Neotropical ticks for their maintenance. The observation of same genovariants in natural hosts and their associated tick species, as capybaras and *Am. romittii* or passerines and *Am. longirostre*, highlights the specific role played by wildlife in the spread of Neotropical *Ehrlichia* and *Anaplasma* infections. However, these infections are not strictly limited to wildlife and may further affect domestic animals as recently exemplified by the detection of an *Anaplasma* usually detected in sloths, *Ca.* Anaplasma amazonensis, in domestic cats in Brazil[42]. The diagnosis of an infection by a related species, *Ca.* Anaplasma sparouinense, in a gold miner also indicates that these Neotropical sylvatic infections may affect humans, probably through the bite of infected ticks[20]. In the Amazon biome, logging, illegal gold mining and human encroachments, agricultural practices, and infrastructure development have created an unprecedented promiscuity with wildlife[31]. This context provides opportunities for the emergence of zoonotic diseases[31], and may promote the spread of *Ehrlichia* and *Anaplasma* infections from their primary natural hosts to humans and domestic animals. In addition, the detection of *Ca.* Allocryptoplasma in French Guiana, as well as its recent detection in diverse ticks and vertebrates worldwide[32,43], suggest that it could be an additional emerging, but currently neglected, genus of tick-borne pathogens.

The genomes of Neotropical *Ehrlichia* and *Anaplasma* species encode homologs of virulence factors, including outer membrane proteins, type IV secretion system apparatus, and associated effector proteins, ANK proteins and TPR. Indeed, *Ca.* Anaplasma sparouinense, *Ca.* Anaplasma amazonensis, and *Ca.* Ehrlichia cajennense MAGs encode paralogs of MSP2/P44 (*Anaplasma*) and OMP1/P28 (*Ehrlichia*) family immunogenic outer membrane proteins used for generating of surface antigen diversity and enabling immune evasion[7,24,28,44]. The three MAGs also contain T1SS and T4SS secretion systems and genes encoding homologs for effector proteins as ANK that are used by *E. chaffeensis* and *A. phagocytophilum* to bind to mammalian cell machinery and modulate its function[7,24,28,44]. Overall, these genomic features suggest that *Ca.* Anaplasma sparouinense, *Ca.* Anaplasma amazonensis, and *Ca.* Ehrlichia cajennense might use similar mechanisms to other *Ehrlichia* and *Anaplasma* species to enter and infect mammalian cells. An important feature is the absence of TRP in the *Ca.* Ehrlichia cajennense MAG whereas these proteins are highly immunogenic in infected patients[24]. However, a recent mutagenesis study showed that TRPs are not essential for the survival and infection of all *Ehrlichia* species, suggesting that these proteins may not regulate host cell signaling in favor of ehrlichial infection or intracellular pathogenicity[45].

In conclusion, this study reveals that *Anaplasma* spp., *Ehrlichia* spp., and, at a lesser extent, *Ca.* Allocryptoplasma, are widespread and diverse in rainforest wildlife and associated ticks in French Guiana. This observation is not limited to Anaplasmataceae, as recent surveys in French Guiana have also revealed substantial diversity for other tick-borne pathogens[46–49]. It corroborates the observation that the high biodiversity of pathogens mirrors the biodiversity level of vertebrates in rainforests[31]. This underlines the need to better document the diversity of tick-borne pathogens in wildlife, particularly in regions where human settlements are increasing rapidly.

## Methods

### Collection of samples

A collection of archived individual DNA templates extracted from 1919 specimens was used (one DNA template per specimen was examined; Source Data). DNA was extracted either from blood samples (for humans, wild mammals, and passerines), spleen samples (wild mammals), or whole body (ticks) of specimens collected from 1994 to 2021 (Source Data). We had primarily collected these samples as part of project evaluation studies among human populations, wildlife, and ticks in French Guiana (See methodology in[20,35,46,47,50,51]. Human blood samples were collected from illegal gold miners working deep in the French Guianese rainforest[51]. Because of the remoteness of their mining camps and their illegal administrative situation, the blood sampling was implemented at 'resting sites', which are transborder areas located in Surinam, on its eastern border with French Guiana[51]. Gold miners go to these informal settlements for transactions of gold and logistical supplies. A previous study along the borders of French Guiana demonstrated that these sites are strategic to efficiently target this highly mobile population for public health actions[52].

### Ethics and regulation

Human samples were collected in Suriname with the approval of the National Ethics Board of Suriname (CMWO (Commissie voor Mensgebonden Wetenschappelijk Onderzoek), Opinion Number VG 25-17). Written informed consent was obtained from human participants. The authorization of importation of human biological samples to France was obtained and the biological collection declared to the French Ministry of Education and Research (DC-2021-4649). The database was anonymized and registered to the Data Protection Officer according to the General Data Protection Regulation (GDPR)[51]. Non-human vertebrate samples were collected and used in accordance with an international CITES permit (Convention on International Trade in Endangered Species of Wild Fauna and Flora; permit FR973A) following the French legislation. Sharing policy in French Guiana, non-human mammal samples are registered in the collection JAGUARS (http://kwata.net/la-collection-jaguars-pour-l-etude-de-la-biodiversite.html;

CITES reference: FR973A) supported by Kwata NGO (accredited by the French Ministry of the Environment and the Prefecture of French Guiana, Agreement R03-2019-06-19-13), Institut Pasteur de la Guyane, DEAL Guyane, Collectivité Territoriale de la Guyane, and validated by the French Guianese prefectural decree n°2012/110. French Ministry of Higher Education and Research provides authorization for projects using wild animals for scientific purposes (reference APAFIS-37571-2022111610578451). Permit number for bird sampling (French Guiana prefectural decrees n°2011/003, 2013/127 and R03-2018-10-30-0092) authorized the capture, marking, sampling, holding and transport of bird samples. Bird sampling was also done with permissions from several organizations: the Direction de l'Environnement, de l'Aménagement et du Logement (DEAL) de Guyane, the Direction Régionale de l'Office National des Forêts (ONF) de Guyane, the Conservatoire du Littoral, the Centre National d'Etudes Spatiales (CNES), the Centre Spatial Guyanais (CSG), the Association pour la Découverte de la Nature en Guyane, the association Randoroura. The use of bird genetic resources is declared to the French Ministry of the Environment under reference TREL1820249A/49 in accordance with the Nagoya Protocol on Access and Benefit Sharing (ABS). French Ministry of the Environment validated the collect and use of tick samples under the reference TREL19028117S/156 in accordance with ABS Nagoya Protocol. All animals were handled in strict accordance with good animal practice and ethical standards as defined by the French code of practice for the care and use of animals for scientific purposes, established by articles R214-87 to R214-137 of the French rural code.

## Molecular survey and typing

The presence of Anaplasmataceae bacteria within each DNA template was investigated through high-throughput 16S rDNA (*rrs*) sequencing (bacterial barcoding, in accordance with the molecular and bioinformatic methods described in[50]. To this aim, a 251-bp portion of the V4 variable region of the bacterial 16S rDNA was amplified from each DNA sample. Polymerase chain reactions (PCR) were conducted using a Multiplex PCR Kit (Qiagen). Amplified bacterial 16S rDNA products were purified and sequenced on an Illumina MiSeq platform (GenSeq, University of Montpellier) and 250-bp end sequence reads were obtained. Sequences with 97% similarity were clustered together and identified as an operational taxonomic unit (OTU). Each OTU sequence was aligned and taxonomically assigned using the Silva database (https://www.arb-silva.de/) allowing the detection of Anaplasmataceae bacterial reads. All bioinformatic analyses were conducted using the Frogs pipeline (https://github.com/geraldinepascal/FROGS).

All *Ehrlichia*, *Anaplasma*, and *Ca*. Allocryptoplasma positive samples were further confirmed through independent PCR amplifications using 16S rDNA species-specific primers and, for a subsample of positive samples, using *gltA* species-specific PCR assays (Supplementary Table 5)[53]. PCR products were visualized through electrophoresis in a 1.5% agarose gel. All positive PCR products were next purified and sequenced in both directions with the Sanger technology (Eurofins). Sequence chromatograms were manually cleaned with Chromas Lite (http://www.technelysium.com.au/chromas_lite. html) and aligned with ClustalW[54] implemented in MEGA[55]. Sequences that differed by one or more nucleotides were assigned distinct allele numbers using DNASP[56]. Further statistical analyses were carried out using R.

## Genome sequencing, assembly, and annotation

*Ca*. Anaplasma sparouinense and *Ca*. Anaplasma amazonensis MAGs were obtained from the sequencing of a human blood sample collected in 2019 (Source Data) and a blood sample of a brown-throated three-toed sloth collected in 1994 (Source Data), respectively, using the Illumina HiSeq 2500 technology and after a library preparation performed with Nextera XT. The raw paired-end reads quality was evaluated with FastQC[57] and the reads were further trimmed via

Atropos[58]. We then obtained 217,683,119 paired-end reads from the metagenome of the human blood sample and 197,624,137 paired-end reads from the metagenome of the sloth blood sample. MAG of *Ca*. Ehrlichia cajennense was retrieved from the metagenome of an *Am. cajennense* tick sample collected in 2021 (Source Data), sequenced using the Oxford Nanopore technology and after a library preparation performed with the Ligation Sequencing kit SQK-LSK 108. The library was sequenced on an R9.4 flowcell on the MinION sequencer for 48 h. Basecalling was done using Albacore v2.0.1 using a quality threshold of Q7. After discarding low quality (< Q7), we obtained 1,088,353 reads with an average quality of 11.97 and average length of 417 bp (range: 98.85 bp-59.270 kb; 10% reads > 1,187 bp).

Metagenomes from Illumina paired-end reads were assembled using MEGAHIT (v1.2.9) with default parameters[59]. Contigs were further automatically clustered into genomes using the algorithm Concoct (v1.1.0)[60] coupled with taxonomy tools implemented in the anvi'o pipeline[61]. Assignation of genomes was next confirmed using the online NCBI BLAST tool (https://blast.ncbi.nlm.nih.gov/Blast.cgi). Metagenome from Oxford Nanopore reads was assembled using Flye (v2.4.1-release)[62]. Consensus contig sequences were produced based on this raw assembly and the original reads' dataset using the Medaka tool (v1.5.0) (Oxford Nanopore). The identification of the *Ca*. Ehrlichia cajennense MAG is based on the taxonomic assignation of the 16S rDNA sequence of a circular contig visualized with Bandage (v0.8.1)[63] (assignation based on the online NCBI BLAST tool). The quality assessment and the completeness of each genome were estimated using QUAST (v4.6.3)[64] and miComplete (-hmms Bact105) (v1.1.1)[65]. The newly obtained genomes were annotated using Prokka (v1.13.1)[66] with default parameters. Graphical representations of newly sequenced genomes were produced using CGView (v1.5)[67].

## Genomic content and virulence factor analyses

Pseudogenes prediction was performed using Pseudofinder (v1.0)[68] on each MAG. The genomic data sets were filtered excluding pseudogenes from the subsequent analyses. To compare the genomic content of obtained MAGs and representative species' genomes of each genus (*Ehrlichia* and *Anaplasma*), Single Copy Orthologs (SCOs) were identified using OrthoFinder (v2.3.12)[69], then SCO lists were retrieved on a R environment to build Venn diagrams using the 'VennDiagram' R package[70]. Ankyrin (ANK) domains and porin proteins from the OMP1 superfamily were identified using the NCBI Conserved Domains Database (CDD, https://www.ncbi.nlm.nih.gov/Structure/cdd) and the SMART database (http://smart.embl-heidelberg.de/). A select number of Prokka-translated proteins underwent this process, initiated by a preliminary BLASTp search based on ANK-containing proteins and OMP1 homologs from reference genomes as queries (*Anaplasma phagocytophilum* (strain HZ, CP006617) and *Ehrlichia chaffeensis* (strain West Paces, CP007480)). All designated proteins were subsequently subjected to testing. Genes encoding for other outer membrane proteins, secretion systems, and effector proteins were detected and retrieved using OrthoFinder and a combination of BLASTn, BLASTp, and tBLASTn.

## Phylogenomics

SCOs shared by the obtained MAGs and representative species' genomes of each genus (*Ehrlichia* and *Anaplasma*) were identified using Orthofinder, then aligned with MAFFT (v7.450)[71] and poorly aligned positions were removed using trimAl (v1.2rev59)[72] prior to concatenation of individual alignments with AMAS (v1.01)[73]. Finally, the most suitable evolutionary model was determined using modeltest-ng (v0.1.5)[74] according to the Akaike information criterion and maximum likelihood (ML) trees were inferred using RAxML (v8.2.9)[75] with 1,000 bootstrap replicates. Free silhouette images used in figures were from the PhyloPic database (https://www.phylopic.org/).

**Reporting summary**

Further information on research design is available in the Nature Portfolio Reporting Summary linked to this article.

## Data availability

The 16S rDNA and *gltA* nucleotide sequences generated in this study have been deposited in the GenBank database under accession code OR854269-OR854350 and PP400975-PP400976 (https://www.ncbi.nlm.nih.gov/genbank/). The raw genomic data and metagenome-Assembled Genomes (MAGs) produced and analyzed in our study are available in Genome Sequence Archive (GSA) database under project PRJCA019774 (https://ngdc.cncb.ac.cn/gsa/). Source data on infection distribution and prevalence generated in this study are provided with this paper in the Source Data. Source data are provided with this paper.

## Code availability

All the codes for this study are stored in GitHub (https://github.com/mariebuysse/Anaplasmataceae_ms) under the https://doi.org/10.5281/zenodo.10911528.

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

## Acknowledgements

We acknowledge the platforms MGX (Montpellier Genomics and Bioinformatics Facility) and GenSeq (Montpellier University) for technical help and high-throughput sequencing, and the ISO 9001 certified IRD i-Trop HPC (South Green Platform; www.southgreen.fr) at IRD Montpellier for providing HPC resources that have contributed to the research results reported in this paper. We thank Roxane Barosi, Mylène Cébé, Bruno Faivre, Noor Fattar, Audrey Godin, Aurélie Khimoun, Gilles Leblond, Gwendoline Le Liard, Maxime Loubon, Céline Michaud, Louise Mutricy, Antoine Perrin, Denis Roussel, Renaud Scheifler, Frank Théron, the Centre de Ressources Biologiques Amazonie, and the Groupe d'Etude et de Protection des Oiseaux en Guyane (GEPOG) for their precious help. We also thank the 'Investissements d'Avenir' grants managed by the French Agence Nationale de la Recherche (ANR, France, ref. ANR-21-CE02-0002, and Laboratoire d'Excellence CEBA, ref. ANR-10-LABX-25-01); the 2015–2020 State-Region Planning Contracts (CPER); the European Regional Development Fund (FEDER) of the University of Poitiers (BiodivUP project); the Malakit study funded by the European Union, the Global Fund, Cayenne Hospital and the French Guiana Health Regional Agency; the ERA-Net Net-Biome 2010, the Fundação para a Ciência e a Tecnologia; the Conseil Regional de Guyane; and the Conseil Regional de Bourgogne.

## Author contributions

M.B., R.K., F.B., and O.D. designed the study. M.B., F.B., B.T., X.B., S.G., M.D., C.C., F.D., F.C., and O.D. collected samples, identified species, and prepared DNA collection. M.B., R.K., and F.B. conducted the analyses under the supervision of O.D., and D.B. helped with the genomic analyses. M.B. and O.D. wrote the draft of the manuscript. All authors contributed to and approved the final version of the manuscript.

## Competing interests

The authors declare no competing interests.
