## [Peer Review File · Nature Communications]

Reviewers' comments:

Reviewer #1 (Remarks to the Author):

The manuscript by Buysse et al reports the genome sequences of three Anaplasmataceae. The authors examined nearly 2000 samples and detected 16 genovariants in 18 host species. This is an interesting manuscript and provides worthwhile data to an understudied field.

These authors come at this with powerful tools but relatively little domain knowledge, which is apparent primarily in the introduction. While the diseases caused by these organisms may generally be called anaplasmosis or ehrlichiosis, the way the introduction is written it makes it sound like the different species cause exactly the same disease. The disease called anaplasmosis was so named over 100 years ago by Sir Arnold Theiler and is caused by *Anaplasma marginale*. The disease caused by *A. phagocytophilum* is called human granulocytic anaplasmosis, and occasionally human anaplasmosis. The organisms infect different cell types and the diseases caused by these two organisms are not the same, and to lump them under one generic name implies such. Similarly, *Ehrlichia chaffeensis* causes a disease known as human monocytic ehrlichiosis, *Ehrlichia ewingii* causes human ewingii ehrlichiosis. While the agent of African heartwater, *Ehrlichia ruminantium* infects vascular endothelial cells.

The authors are correct that *Anaplasma capra* is an emerging pathogen in China. It was only recently that investigators examined what cell type is infected by this agent, and their analysis showed that it infects the erythrocyte, ie a different cell type than that of *A. phagocytophilum*, which infects neutrophils. *A. capra* is more closely related to the *A. marginale* clade of Anaplasmas (even in this paper), so to lump it as causing the same disease as *A. phagocytophilum* is incorrect. See: Peng et al., 2021. doi: 10.1080/22221751.2021.1876532

Line 64ff: there is occasional spill over of these animal-infecting species to humans. See

M T E P Allsopp et al., 2005, doi: 10.1196/annals.1355.060... which is really what HGA is. There are several examples of this... but hard to track down.

Line 148ff: In this paragraph, shouldn't you use "genovariant" rather than species (spp)?

The authors indicate that they have 16 genovariants, but, for example genovariant 9 and 10 are 99.65% identical to each other, yet tree quite differently. They both have the same identity to

Ehrlichia sp. VKAA024 of Rhipicephalus microplus, Malaysia (99.91%; KY046297), yet one trees as identical, and the other is in a separate subclade... Is this correct? I was originally going to ask at what point do you call something the same, as some of the genovariants are very similar... 99.65% is ~5 bp different.

Figure 2 and Figure 3 are in microdot. For figure 2, can you just present the hosts that were infected, and the put the uninfected ones in the supplemental? In figure 3, can you not put every identical sample in the figure, but instead indicate the number that was the same? For example "Ehrlichia sp. of Pipra aureola * (7)"

Regarding Figure 5, and the analysis for that figure: *A. marginale* actually has many more virB2 sequences than *A. phagocytophilum* (See Gillespie et al 2010, doi: 10.1128/IAI.01384-09) but they were not annotated as such in the original annotation. You may want to see if the *Ca. A. spar.* sequences match those. *A. marginale* also has a virB7, described in the Gillespie paper. There are 4 Ank domain containing proteins in *A. phagocytophilum*, and 3 in *A. marginale* (SMART is much better at finding Ank domains than CDD). I don't really understand the presentation of the T4SS genes in this random order... why not just put them in numerical order?

You present the OMPs of the Ehrlichial agent in Fig 5, but not of the Anaplasma agents. It would be somewhat unusual (maybe! we don't really know as we only really have the tip of the iceberg with Anaplasma diversity, but, at least amongst the known species) for different species to have identical msp2 sequences. There is much literature on these genes as important factors for immune evasion, and it would be nice to have these covered more thoroughly.... Maybe the genomic organization, can you define the expression site, is it an operon as in other species.

It should be acknowledged that the Anaplasma sequences are not complete, therefore any reference to genes not present could be in a gap. I don't quite understand how you have a "complete" circular genome, for the Ehrlichia, but a completeness score of 90%, while with the uncompleted genomes you find a completeness score of 97%. Also, your coding densities are much higher than typical.

To find only 1 human infection in ~360 samples is not a lot, so to state that this is a health hazard might be overstating things. You might not want to lead with that.

Minor corrections:

Ref 1 is to the US CDC. This statement is applicable to the US. Rickettsioses are likely more frequently diagnosed outside the US. Safer to say something like "Rickettsial infections are the most common

infections after Lyme, and among these rickettsial organisms are Ehrlichia spp. and Anaplasma spp.”
or something like that.

Grammar: “The clinical manifestations of human ehrlichiosis and anaplasmosis are similar, ranging from subclinical to life-threatening diseases associated with multi-organ failure”

Replace diseases with symptoms

Line 52; infections should be organisms

Line 59: heart water is one word

Line 159: delete “the”, and change “have” to “has”

Line 168: I don’t understand latter part of this sentence:

“While the Anaplasma genovariant #12 of the tropical cattle tick, Rhipicephalus microplus, is identical to A. marginale, all other genovariants are distinct to taxa referenced in public database for their 16S rDNA sequences.”

Also – delete first “the, and database should be plural.

Emerging Anaplasmosis and Ehrlichiosis in the Amazon Biome, by Buysse et al. (NCOMMS-23-40521). Response to reviewer

Note: All key modifications we made in the revised version of the manuscript are in track changes mode.

Reviewer #1 (Remarks to the Author):

The manuscript by Buysse et al reports the genome sequences of three Anaplasmataceae. The authors examined nearly 2000 samples and detected 16 genovariants in 18 host species. This is an interesting manuscript and provides worthwhile data to an understudied field.

*These authors come at this with powerful tools but relatively little domain knowledge, which is apparent primarily in the introduction. While the diseases caused by these organisms may generally be called anaplasmosis or ehrlichiosis, the way the introduction is written it makes it sound like the different species cause exactly the same disease. The disease called anaplasmosis was so named over 100 years ago by Sir Arnold Theiler and is caused by *Anaplasma marginale*. The disease caused by *A. phagocytophilum* is called human granulocytic anaplasmosis, and occasionally human anaplasmosis. The organisms infect different cell types and the diseases caused by these two organisms are not the same, and to lump them under one generic name implies such. Similarly, *Ehrlichia chaffeensis* causes a disease known as human monocytic ehrlichiosis, *Ehrlichia ewingii* causes human ewingii ehrlichiosis. While the agent of African heartwater, *Ehrlichia ruminantium* infects vascular endothelial cells. The authors are correct that *Anaplasma capra* is an emerging pathogen in China. It was only recently that investigators examined what cell type is infected by this agent, and their analysis showed that it infects the erythrocyte, ie a different cell type than that of *A. phagocytophilum*, which infects neutrophils. *A. capra* is more closely related to the *A. marginale* clade of Anaplasmas (even in this paper), so to lump it as causing the same disease as *A. phagocytophilum* is incorrect. See: Peng et al., 2021. doi: 10.1080/22221751.2021.1876532*

> Reply #1: We express our gratitude to the reviewer for their insightful comments and the wealth of expertise they provided. Acknowledging the importance of the reviewer's observation, we have revised our introductory paragraph to provide a more detailed presentation of these diseases. We now explicitly highlight the differences between the forms of anaplasmoses and ehrlichioses (lines 41-55 and 48-51). Ultimately, we regret if the reviewer perceived our presentation of ehrlichioses and anaplasmoses in the introduction as inadequate. It is more of a misunderstanding than an actual substantive issue.

We acknowledge that our first part of the introduction may be considered too brief. In consideration of the concise format required by *Nature Communications*, we deliberately chose to introduce both ehrlichioses and anaplasmoses together, necessitating certain shortcuts. Most articles on these diseases generally focus on a single one, such as bovine anaplasmosis, usually providing a detailed history of the specific disease. This is not the case for us: By concentrating on ehrlichioses and anaplasmoses as

a whole, we cannot delve into the details of each individual one. Nevertheless, we fully agree with the reviewer that different ehrlichioses and anaplasmoses are distinct diseases. At no point did we intend to imply that these are the same diseases, although they share evident commonalities: The causative agents target various blood cells (erythrocytes, granulocytes, platelets, vascular endothelial cells, etc.), and clinical symptoms are non-specific (including an acute onset of fever, headache, myalgia, and malaise). It is precisely these commonalities that we aimed to emphasize in the first version of our manuscript.

Line 64ff: there is occasional spill over of these animal-infecting species to humans. See M T E P Allsopp et al., 2005, doi: 10.1196/annals.1355.060... which is really what HGA is. There are several examples of this... but hard to track down.

> **Reply #2:** Certainly, and we wholeheartedly agree with the reviewer. This is indeed applicable to species of *Anaplasma* and *Ehrlichia* infecting human populations: It occurs when these pathogens (all of animal origin) are transmitted to humans through tick bites. This precisely aligns with the definition of a spill-over. We have added a description of this process to the introduction and the discussion (lines 68-69 and 303-305).

Line 148ff: In this paragraph, shouldn't you use "genovariant" rather than species (spp)?

> **Reply #3:** Yes, that is indeed more accurate, thank you. We have replaced "species" with "genovariant" (lines 158-166).

The authors indicate that they have 16 genovariants, but, for example genovariant 9 and 10 are 99.65% identical to each other, yet tree quite differently. They both have the same identity to Ehrlichia sp. VKAA024 of Rhipicephalus microplus, Malaysia (99.91%; KY046297), yet one trees as identical, and the other is in a separate subclade... Is this correct? I was originally going to ask at what point do you call something the same, as some of the genovariants are very similar... 99.65% is ~5 bp different.

> **Reply #4:** It is indeed exact that we observed 16 genovariants, and their pairwise nucleotide identities vary depending on the genovariants considered. While some genovariants are highly similar (i.e., with a strong pairwise nucleotide identity), others are markedly different (i.e., with a low pairwise nucleotide identity). This is precisely what we describe in the results (lines 148-194, spanning five consecutive paragraphs) and in Supplementary Table 2.

In simpler terms, if two bacteria share the same 16S rDNA sequence, we considered them the same genovariant. However, if they differed by even a single nucleotide, we classified them as different genovariants. We strictly adhered to the definition of a genovariant as presented in the text. To further clarify this crucial point, we have added an additional sentence (line 153-156).

Regarding the interpretation of the 16S rDNA phylogenetic tree (Figure 3), it is actually obvious that *Ehrlichia* genovariants 9 and 10 are indeed very close and cluster as sister branches, as expected given their strong pairwise nucleotide identity. The reviewer can easily confirm this by examining the scale bar of this phylogeny: The branch length separating *Ehrlichia* genovariants 9 and 10 corresponds well to a strong pairwise nucleotide identity, consistent with the details provided in Supplementary Table 2.

*Figure 2 and Figure 3 are in microdot. For figure 2, can you just present the hosts that were infected, and the put the uninfected ones in the supplemental? In figure 3, can you not put every identical sample in the figure, but instead indicate the number that was the same? For example “Ehrlichia sp. of Pipra aureola * (7)”*

> Reply #5: Certainly, we have also considered this aspect prior to drafting our manuscript. We arbitrarily opted to include all details to allow reviewers to verify the complexities of our results. Until the appeals process is concluded, we prefer to maintain them in this comprehensive form, as despite their apparent size, they offer greater precision. At the conclusion of the appeals process, if the editors deem it necessary, we are open to refining these figures and making them more concise.

Regarding Figure 5, and the analysis for that figure: A. marginale actually has many more virB2 sequences than A. phagocytophilum (See Gillespie et al 2010, doi: 10.1128/IAI.01384-09) but they were not annotated as such in the original annotation. You may want to see if the Ca. A. spar. sequences match those. A. marginale also has a virB7, described in the Gillespie paper. There are 4 Ank domain containing proteins in A. phagocytophilum, and 3 in A. marginale (SMART is much better at finding Ank domains than CDD). I don't really understand the presentation of the T4SS genes in this random order... why not just put them in numerical order?

> Reply #6: Yes, that's accurate. Initially, we relied on the original annotation of the *A. marginale* genome. Subsequently, all additional virB2, virB7, and Ank sequences in *A. marginale* and *A. phagocytophilum*, as highlighted by the reviewer, are now included in the revised Figure 5A. We addressed this concern by incorporating these sequences into our novel bioinformatic analysis. However, no additional virB2, virB7, and Ank sequences were detected in our MAGs beyond those previously listed in Figure 5A. Consequently, the inclusion of these additional sequences does not alter our conclusions: *Ca. Anaplasma sparouinense*, *Ca. Anaplasma amazonensis*, and *Ca. Ehrlichia cajennense* possess most of the virulence factors identified in the pathogenic *Ehrlichia* and *Anaplasma* species for humans and domestic animals.

Following the reviewer's recommendations, we have also arranged the T4SS genes in numerical order (see revised Figure 5A).

You present the OMPs of the Ehrlichial agent in Fig 5, but not of the Anaplasma agents. It would be somewhat unusual (maybe! we don't really know as we only really have the tip of the iceberg with

Anaplasma diversity, but, at least amongst the known species) for different species to have identical msp2 sequences. There is much literature on these genes as important factors for immune evasion, and it would be nice to have these covered more thoroughly.... Maybe the genomic organization, can you define the expression site, is it an operon as in other species.

> Reply #7: We chose to exclusively represent the outer membrane proteins (OMPs) structure of *Ehrlichia chaffeensis* and *Ca. Ehrlichia cajennense*. The primary factor guiding this decision is the fragmentation of the retrieved genomes. While we obtained a circular (single, uninterrupted contig) *Ca. Ehrlichia cajennense* genome, both *Ca. Anaplasma* genomes are fragmented. Our analysis of OMP presence revealed that their sequences are dispersed across different contigs. The fragmented nature of the genomes prevents a meaningful discussion of their structural organization. Moreover, representing the positioning of OMPs for *Anaplasma* spp., as we have done for *Ehrlichia* spp., is not deemed relevant. In *Anaplasma* spp., most OMPs are not structured within a large cluster, as observed in *Ehrlichia* spp.; instead, they are scattered along the genomes. This distribution pattern is also evident in *Ca. Anaplasma sparouinense* and *Ca. Anaplasma amazonensis*, where OMPs were detected on several distinct contigs. This crucial observation is now detailed in the revised results (lines 269-274).

It should be acknowledged that the Anaplasma sequences are not complete, therefore any reference to genes not present could be in a gap. I don't quite understand how you have a "complete" circular genome, for the Ehrlichia, but a completeness score of 90%, while with the uncompleted genomes you find a completeness score of 97%. Also, your coding densities are much higher than typical.

> Reply #8: Yes, it's true that our genomic sequences of *Anaplasma* are not complete, and we explicitly state this in the results (lines 219-223) and Supplementary Table S3, along with all other statistics from our metagenomes. However, the completeness levels are actually excellent for a metagenomic approach: all are above 90%, and even above 97% for *Ca. Anaplasma sparouinense* and *Ca. Anaplasma amazonensis*. We have taken this into consideration, particularly examining the possibility that any gene not present could be in a gap. We are confident that this is not the case, as notably illustrated in Figures 4B and 4C. For example, there are only 13 genes not found in *Ca. Anaplasma sparouinense* and *Ca. Anaplasma amazonensis* that are present in *A. marginale*, *A. platys*, and *A. phagocytophilum*. This is a value very similar to what is observed among known *Anaplasma* species (see Figure 4B for example). Conversely, we have detected genes present in *Ca. Anaplasma sparouinense* and *Ca. Anaplasma amazonensis* but absent in *A. marginale*, *A. platys*, and *A. phagocytophilum*. All of this suggests that our analysis has been rigorous; however, as suggested by the reviewer, we discuss the possibility that a gene not present could be in a gap (lines 223-228).

Regarding the complete circular genome of *Ehrlichia*, we would like to note here that it was obtained through hybrid sequencing (short and long reads), and the reconstructed genome was unambiguously circular and thus complete. The completeness score of 90% simply reflects the substantial variation in

gene content among *Ehrlichia* species (much more than among *Anaplasma* species). We have added two short sentences to explain this score within the revised manuscript (lines 230-233).

To find only 1 human infection in ~360 samples is not a lot, so to state that this is a health hazard might be overstating things. You might not want to lead with that.

> **Reply #9:** We have moderated this statement in response to the reviewer's comments (lines 28, 35-36, 301 and 307). However, we respectfully disagree with the essence of this comment. In comparison, in 2019, there was a notable peak of 5,655 cases of human granulocytic anaplasmosis in the USA (making it the most prevalent anaplasmosis in humans) according to the CDC (<https://www.cdc.gov/anaplasmosis/stats/index.html>). The population of the USA in that year was 328.3 million, resulting in one human case for >58,000 inhabitants. This figure is significantly lower than what we observe in French Guiana, where we document one case for 360 individuals.

Minor corrections:

Ref 1 is to the US CDC. This statement is applicable to the US. Rickettsioses are likely more frequently diagnosed outside the US. Safer to say something like "Rickettsial infections are the most common infections after Lyme, and among these rickettsial organisms are Ehrlichia spp. and Anaplasma spp." or something like that.

> **Reply #10:** The reviewer is correct, and we have revised the sentence in accordance with its suggestion (lines 39-40).

Grammar: "The clinical manifestations of human ehrlichiosis and anaplasmosis are similar, ranging from subclinical to life-threatening diseases associated with multi-organ failure"

Replace diseases with symptoms

Line 52; infections should be organisms

Line 59: heart water is one word

Line 159: delete "the", and change "have" to "has"

Line 168: I don't understand latter part of this sentence:

"While the Anaplasma genovariant #12 of the tropical cattle tick, Rhipicephalus microplus, is identical to A. marginale, all other genovariants are distinct to taxa referenced in public database for their 16S rDNA sequences."

Also – delete first "the, and database should be plural

> **Reply #11:** All the points have been corrected.

REVIEWER COMMENTS

Reviewer #1 (Remarks to the Author):

I appreciate the author's attention and explanations to my previous comments.

I am disappointed that they did not choose to modify Figure 2 and 3, however, as they suggest, I will leave this decision to the editors. (I understand Zoom features are possible! :-))

I respectfully disagree with the authors about Anaplasma OMPs not being relevant (comment 7). Msp2 is transcribed from a single operon that is syntenic in the Anaplasma genomes. While the distribution of paralogs may not be possible from fragmented genomes, the representation of the operon is relevant and important. Although I do appreciate the revised text re msp2.

Reply 9: This is a false argument. To say that HGA in the entire US was 1 in 58,000 does not equate to 1 in the subset of 360 in this study. The 5655 cases in the US were sick people that were detected by doctors. Your study would more accurately be one case in 286,000; the population of French Guiana. Nevertheless, thank you for moderating the original statement.

Reviewer #3 (Remarks to the Author):

General Comments

The present work aimed to investigate the diversity of Ehrlichia/Anaplasma/Allochromoplasma in wild animals, humans and ticks in the Amazon Rainforest of French Guiana. The results presented are novel, despite some limitations, mainly when it comes to additional molecular characterization of the detected genovariants using other molecular markers (e.g. dsb, groEL, gltA, sodB, ITS, etc.)

I would recommend modification of the title, since authors neither can state that all the genovariants detected are able to infect humans and/or domestic animals, nor the ability to cause diseases (collectively called anaplasmosis and ehrlichiosis). I would rather say: Potential novel Anaplasmataceae agents in humans, wild animals, and ticks in the French Guiana Amazon Rainforest.

Specific Comments

- Line 66: Has *A. marginale* already been detected in humans? Please, double check that. Where? It is cited in which reference?

Anaplasma platys has already been detected in humans: Co-infection with *Anaplasma platys*, *Bartonella henselae* and *Candidatus Mycoplasma haematoparvum* in a veterinarian.

Maggi RG, Mascarelli PE, Havenga LN, Naidoo V, Breitschwerdt EB.

Parasit Vectors. 2013 Apr 15;6:103.

Intravascular persistence of *Anaplasma platys*, *Ehrlichia chaffeensis*, and *Ehrlichia ewingii* DNA in the blood of a dog and two family members.

Breitschwerdt EB, Hegarty BC, Quorollo BA, Saito TB, Maggi RG, Blanton LS, Bouyer DH.

Parasit Vectors. 2014 Jul 1;7:298. doi: 10.1186/1756-3305-7-298.

Anaplasma bovis in humans:

Anaplasma bovis Infection in Fever and Thrombocytopenia Patients - Anhui Province, China, 2021.

Lu M, Chen Q, Qin X, Lyu Y, Teng Z, Li K, Yu L, Jin X, Chang H, Wang W, Hong D, Sun Y, Kan B, Gong L, Qin T.

China CDC Wkly. 2022 Mar 25;4(12):249-253.

What about infection by *Anaplasma ovis* in humans?

- Line 73: Rephrase as following: In 2022, a chronic infection by an intraerythrocytic *Anaplasma* sp. was diagnosed..

- Line 96: POTENTIAL new hazards

- Authors should acknowledge limitations of the present, such as the lack of additional molecular characterization based on distinct molecular markers (e.h. dsb, gltA for Ehrlichia; groEL, 23S-5S for Anaplasma), in order to phylogenetically position the novel genotypes with those previously detected in wild animals from South America, specially Brazil.

For instance: the Ehrlichia genotypes detected in Didelphis would be closely related to those found in opossums from Brazil?

Please, check it out the following MS:

RAGA, MARIA DO SOCORRO COSTA OLIVEIRA ; COSTA, FRANCISCO BORGES ; CALCHI, ANA CLÁUDIA ; DE MELLO, VICTÓRIA VALENTE CALIFRE ; MONGRUEL, ANNA CLAUDIA BAUMEL ; DIAS, CLARA MORATO ; BASSINI-SILVA, RICARDO ; SILVA, ELLAINY MARIA CONCEIÇÃO ; PEREIRA, JOSÉ GOMES ; RIBEIRO, LARISSA SARMENTO DOS SANTOS ; DA COSTA, ANDRÉA PEREIRA ; DE ANDRADE, FABIO HENRIQUE EVANGELISTA ; SILVA, ANA LUCIA ABREU ; Machado, Rosangela Zacarias ; André, Marcos Rogério . Molecular detection and characterization of vector-borne agents in common opossums (*Didelphis marsupialis*) from northeastern Brazil. ACTA TROPICA, v. 244, p. 106955, 2023.

André, Marcos Rogério; CALCHI, ANA CLÁUDIA ; PERLES, LIVIA ; GONÇALVES, LUIZ RICARDO ; UCCELLA, LUCAS ; LEMES, JHESSYE RAFAELA BATISTA ; NANTES, WESLEY ARRUDA GIMENES ; SANTOS, FILIPE MARTINS ; PORFÍRIO, GRASIELA EDITH DE OLIVEIRA ; BARROS-BATTESTI, DARCI MORAES ; HERRERA, HEITOR MIRAGLIA ; Machado, Rosangela Zacarias . Novel Ehrlichia and Hepatozoon genotypes in white-eared opossums (*Didelphis albiventris*) and associated ticks from Brazil. Ticks and Tick-Borne Diseases, v. 13, p. 102022, 2022.

Was not the genovariant Ehrlichia #6 similar to Candidatus Ehrlichia dumleri? Please, perform another nBLAST search to double check that, since Ca. Ehrlichia dumleri was closely related to an Ehrlichia genotype detect in *E. barbara* from Brazil.

What about the genovariant detected in capybaras? Was not it similar to the Candidatus Ehrlichia capybara previously detected in Brazil? Please, check it out: Novel Anaplasmataceae agents Candidatus Ehrlichia hydrochoerus and Anaplasma spp. Infecting Capybaras, Brazil

- Line 179: Did not the authors find high identity to Ehrlichia minasensis in *R. microplus* ticks?

- Line 199: coelebs

- Lines 304-305: Regarding the sentence: "The ability of Ca. Anaplasma sparouinense to cause intraerythrocytic anaplasmosis in humans shows that there are spillovers from wildlife in Amazon

rainforests.” Authors cannot state that, since this genovariant has not been detected in wild animals so far. Please, rephrase or delete this sentence. [L
SEP]

- Line 311: Please, remove “Amazonian”, since not all the articles cited in this sentence referred to animals sampled in Amazon only.

- Line 315: What about the genovariants detected in capybaras? Did they differ from those detected in capybaras from Brazil and Argentina?

- Line 319: Be careful when generalizing new genovariants in Brazil as all belonging to Amazon Rainforest!

E.g. Candidatus Anaplasma brasiliensis was detected in Cerrado biome but not in the Amazon biome.

I strongly recommend authors check the biomes where these novel genotypes were described in Brazil.

- Line 689: shows

- Supplementary Table 2: Please, include the size (bp) of the sequences obtained for each genovariant, as well as query cover and E-value. Please, indicate the date when nBLAST search was performed.

- Table 3: What N50 and L50 stand for?

Response to reviewers

Novel *Anaplasma* and *Ehrlichia* bacteria in humans, wildlife, and ticks in the Amazon rainforest, by Buysse et al. (formerly 'Emerging Anaplasmosis and Ehrlichiosis in the Amazon Biome': NCOMMS-23-40521).

Note: All key modifications we made in the revised version of the manuscript are in track changes mode.

Reviewer #1 (Remarks to the Author):

I appreciate the author's attention and explanations to my previous comments.

I am disappointed that they did not choose to modify Figure 2 and 3, however, as they suggest, I will leave this decision to the editors. (I understand Zoom features are possible! :-))

> Reply #1: We would like to sincerely thank the reviewer for agreeing to reevaluate our study. Her/His valuable advice has significantly contributed to advancing and improving our manuscript. Thank you for presenting us with this challenge ;-)

We have addressed the issue with figures by modifying Figures 2 and 3 based on the suggestions provided by the reviewer. In Figure 2, we now only present the hosts that were infected and modified the Figure legend accordingly (lines 764-766). The complete version of Figure 2, including both infected and uninfected hosts (the original version of Figure 2), is now included in the Supplementary Information file and referred to as Supplementary Figure 1 in the text. Similarly, in Figure 3, we now display only one sequence per genovariant and per host species on the phylogenetic tree. The Figure legend has been modified accordingly (lines 775-778). As with Figure 2, the complete version of Figure 3 is now presented in the Supplementary Information file and referred to as Supplementary Figure 2 in the text. The new versions of Figures 2 and 3 are therefore simpler and visually lighter - and the total amount of information is still preserved and presented in the new Supplementary Figure 1 and 2.

*I respectfully disagree with the authors about *Anaplasma* OMPs not being relevant (comment 7). *Msp2* is transcribed from a single operon that is syntenic in the *Anaplasma* genomes. While the distribution of paralogs may not be possible from fragmented genomes, the representation of the operon is relevant and important. Although I do appreciate the revised text re *msp2*.*

> Reply #2: We understand the reviewer's perspective and have produced a schematic figure illustrating the structure of the operon. Although genome fragmentation hinders the reconstruction of the *msp2/p44* operon structure, it is probable that *msp2/p44* is transcribed from a single operon. This assertion is supported by the presence of single copies of the transcriptional regulator 1 (*tr1*) and three

operon-associated genes (*OpAG4*, *OpAG3* and *OpAG2*). In *Ca. Anaplasma sparouinense* and *Ca. Anaplasma amazonensis* genomes, an organization of *tr1*, *OpAG4*, *OpAG3* and *OpAG2* loci similar to that observed in *A. marginale* genomes is evident. However, they also exhibit characteristic features of the *A. phagocytophilum msp2/p44* operon, such as a nucleoside diphosphate kinase (*ndK*) gene downstream of *tr1* and the association of one *msp2/p44* copy with the outer-membrane protein 1X (*omp1X*) gene.

As previously mentioned, the difficulty arises from the fragmentation of our genomes and the repetitive nature of *msp2/p44* sequences within these genomes: Some of our contigs start or end precisely with *msp2/p44* sequences (not surprisingly, assembly tools struggle to concatenate contigs when they are bounded by repeated sequences). However, for each of our *Anaplasma* genomes, we have only one contig that ends with *tr1*, *OpAG4*, *OpAG3* and *OpAG2* sequences, and several contigs containing *msp2/p44* (where *msp2/p44* sequences are often found at the ends of contigs). The detection of a single contig that ends with the *tr1*, *OpAG4*, *OpAG3* and *OpAG2* sequences per genome shows that each *Anaplasma* genome harbors a single *msp2/p44* expression locus. The presence of multiple copies of *msp2/p44* is, of course, expected as they serve as donors for recombination at the *msp2/p44* located within the operon. We explain all of this in the revised version of our manuscript (lines 309-322) and have added the schematic figure illustrating the structure of the operon to the supplementary materials (Supplementary Figure 4).

Reply 9: This is a false argument. To say that HGA in the entire US was 1 in 58,000 does not equate to 1 in the subset of 360 in this study. The 5655 cases in the US were sick people that were detected by doctors. Your study would more accurately be one case in 286,000; the population of French Guiana. Nevertheless, thank you for moderating the original statement.

> **Reply #3:** It's a fair point, and the reviewer's perspective is also valid. Fortunately, this issue no longer appears in the current version of the manuscript, as noted by the reviewer.

Reviewer #2 (Remarks to the Author):

General Comments

The present work aimed to investigate the diversity of Ehrlichia/Anaplasma/Alloccryptoplasma in wild animals, humans and ticks in the Amazon Rainforest of French Guiana. The results presented are novel, despite some limitations, mainly when it comes to additional molecular characterization of the detected genovariants using other molecular markers (e.g. dsb, groEL, gltA, sodB, ITS, etc.).

> **Reply #4:** Thank you very much for providing such detailed feedback, both here and below. We have thoroughly considered all of these comments, including those concerning the use of other markers. We address each of them point by point in the following replies (see replies #11, 12, 13, 14,

and 18 below). In the revised version of the manuscript, we have included additional (albeit partial) data on molecular typing, which we produced in the course of this study but never published. All of these novel sequences have been deposited in GenBank, as indicated in lines 526-527. The incorporation of these additional data was pivotal in addressing specific questions regarding similarities with genovariants detected in Brazil, as we detail in subsequent replies. We also took the opportunity to update Supplementary Table 2 (BLASTn against GenBank) following the reviewer's comments; it was a beneficial input. Please note that these additions do not alter the conclusions of our study: Molecular typing identifies *Anaplasma* and *Ehrlichia* infections found in this study as highly endemic, with a majority of new strains and putative species specific to French Guiana. A few of these strains are also found in Brazil, as detailed below.

Furthermore, all of the studies, including those reporting on infections in wildlife in Brazil, pointed out by the reviewer, have been included in the revised version of the manuscript, as we detail below.

I would recommend modification of the title, since authors neither can state that all the genovariants detected are able to infect humans and/or domestic animals, nor the ability to cause diseases (collectively called anaplasmosis and ehrlichiosis). I would rather say: Potential novel Anaplasmataceae agents in humans, wild animals, and ticks in the French Guiana Amazon Rainforest.

> Reply #5: Acknowledging the importance of the reviewer's observation, we have modified the title as follows: 'Novel *Anaplasma* and *Ehrlichia* bacteria in humans, wildlife, and ticks in the Amazon rainforest'. The revised title is more neutral, as it does not imply infectivity for humans and animals, as suggested by the reviewer.

The reviewer's suggestion is justified in the sense that our study aims to explore the genetic and genomic diversity of *Anaplasma* and *Ehrlichia* bacteria, rather than describing clinical facts. However, it is worth noting that all well-known *Anaplasma* and *Ehrlichia* species have been associated in the past with disease, either in humans or animals, without exception. Additionally, some of the genovariants we found have already been associated as causative agents of diseases, such as *A. marginale* or *Candidatus Anaplasma sparouinense*. Therefore, it would be surprising if this will be not the case for the other novel genovariants we found here.

Specific Comments

- Line 66: Has A. marginale already been detected in humans? Please, double check that. Where? It is cited in which reference?

> Reply #6: *A. marginale* has never been detected in humans - the reviewer's concern is entirely valid. We recognize that a portion of this sentence may have been ambiguous and could imply that this

species can infect humans. This was certainly not our intention, and we apologize. Therefore, we have reworded this sentence text to eliminate any ambiguity. It now reads as follows:

‘Most of the other *Ehrlichia* and *Anaplasma* species are major agents of veterinary diseases as exemplified by *E. ruminantium* (causing heartwater, or cowdriosis), *A. marginale* (bovine anaplasmosis), which together are the most common and lethal tick-borne pathogens in cattle, and *E. canis* (canine ehrlichiosis), which is common in dogs in tropical and subtropical regions.’ (lines 63-64).

Anaplasma platys has already been detected in humans:

- *Co-infection with Anaplasma platys, Bartonella henselae and Candidatus Mycoplasma haematoparvum in a veterinarian. Maggi RG, Mascarelli PE, Havenga LN, Naidoo V, Breitschwerdt EB. Parasit Vectors. 2013 Apr 15;6:103.*
- *Intravascular persistence of Anaplasma platys, Ehrlichia chaffeensis, and Ehrlichia ewingii DNA in the blood of a dog and two family members. Breitschwerdt EB, Hegarty BC, Quorollo BA, Saito TB, Maggi RG, Blanton LS, Bouyer DH. Parasit Vectors. 2014 Jul 1;7:298. doi: 10.1186/1756-3305-7-298.*

Anaplasma bovis in humans:

- *Anaplasma bovis Infection in Fever and Thrombocytopenia Patients - Anhui Province, China, 2021. Lu M, Chen Q, Qin X, Lyu Y, Teng Z, Li K, Yu L, Jin X, Chang H, Wang W, Hong D, Sun Y, Kan B, Gong L, Qin T. China CDC Wkly. 2022 Mar 25;4(12):249-253.*

> Reply #7: Certainly, we have also taken into account these publications while drafting our manuscript. We made a deliberate decision not to include them in our list of citations, as we have already referenced a significant number of studies, and we chose not to give particular emphasis to *A. platys* and *A. bovis* over others. However, we acknowledge the reviewer's point that our introduction would be more comprehensive if these references were included. Therefore, we greatly appreciate this suggestion to provide a more exhaustive introduction, and we have now incorporated these three references (lines 67-69).

What about infection by Anaplasma ovis in humans?

> Reply #8: Another valid point, thank you. A single case of *A. ovis* infection in humans has been documented in Cyprus. Following a tick bite, a patient experienced an 11-day fever, moderate anemia, thrombocytopenia, and elevated levels of transaminases. The patient achieved complete recovery after receiving antibiotic treatment (Chochlakis et al., 2010; DOI: 10.3201/eid1606.090175). This reference has been added to the list provided in reply #7 concerning *A. platys* and *A. bovis* (line 69).

- Line 73: *Rephrase as following: In 2022, a chronic infection by an intraerythrocytic Anaplasma sp. was diagnosed.*

> **Reply #9:** Done (lines 73-74).

- Line 96: *POTENTIAL new hazards*

> **Reply #10:** Done: 'potential' is now included in the sentence (line 97).

- *Authors should acknowledge limitations of the present, such as the lack of additional molecular characterization based on distinct molecular markers (e.h. dsb, gltA for Ehrlichia; groeL, 23S-5S for Anaplasma), in order to phylogenetically position the novel genotypes with those previously detected in wild animals from South America, especially Brazil. For instance: the Ehrlichia genotypes detected in Didelphis would be closely related to those found in opossums from Brazil?*

Please, check it out the following MS:

- *Braga, Maria do Socorro Costa Oliveira et al. Molecular detection and characterization of vector-borne agents in common opossums (Didelphis marsupialis) from northeastern Brazil. ACTA TROPICA, v. 244, p. 106955, 2023.*
- *André, Marcos Rogério et al. Novel Ehrlichia and Hepatozoon genotypes in white-eared opossums (Didelphis albiventris) and associated ticks from Brazil. Ticks and Tick-Borne Diseases, v. 13, p. 102022, 2022.*

> **Reply #11:** We would like to express our gratitude to the reviewer for raising this point, as it holds significant importance for our research. We consistently prioritize accuracy in the identification of the bacteria we detect. While our genovariants were initially identified based on the 16S rRNA sequences, it's worth noting that the 16S rRNA gene sequences, although widely used as a bacterial taxonomic marker, often lack the resolution necessary for precise intrageneric phylogenetic analysis. This inadequacy is primarily due to limited sequence polymorphism, especially when utilizing short sequences (approximately 300-400bp, as commonly done in many studies). In light of this limitation, we opted to sequence nearly complete 16S rRNA gene sequences, spanning over 1,500bp, which provides us with more informative genetic data enabling finer distinctions between genovariants. Additionally, we generated complete whole genome sequences for three of these genovariants, further confirming their genetic distinctiveness. From this perspective, the interest of complete 16S rRNA gene sequences and whole genome sequences is obvious for accurate typing purposes.

However, the reviewer is absolutely correct in identifying a crucial point that warrants our attention and careful consideration. When it comes to identifying *Ehrlichia* of opossum, we cannot directly

compare our genetic data with that generated by Braga et al. (2023). They utilized the *dsb* gene as a unique marker, a practice that differs from the standard methodologies employed by most laboratories, including ours, for *Ehrlichia* studies. However, André et al. (2022) employed reliable methods using other genetic markers for typing *Ehrlichia* of opossum, although for another species, the white-eared opossum (*Didelphis albiventris*). We have already generated some gene sequences from our samples that we can compare with data produced by André et al. (2022). Interestingly, one of the short 16S rRNA gene fragments of opossum *Ehrlichia* (316bp, GenBank OK605040) produced by André et al. (2022) is 100% identical to the *Ehrlichia* genovariant #2 we found in French Guiana (also in opossums), suggesting that it could be the same genovariant circulating in opossums in French Guiana and Brazil. We also produced the *gltA* gene sequence of *Ehrlichia* genovariant #2 and found that it was also 100% identical to the Brazilian opossum *Ehrlichia* (GenBank OK763036-OK763038) first identified by André et al. (2022). We will now include this information in the results section of the manuscript (lines 193-202). We have also included in the revised version of the manuscript the *gltA* protocol we used (lines 458-459, Supplementary Table 6), and we have further submitted the *gltA* sequence to GenBank (as mentioned at lines 526-527). The two studies listed by the reviewer, Braga et al. (2023) and André et al. (2022), are now cited in the manuscript.

Additionally, there were potential issues with a few other genovariants, as highlighted by the reviewer, for which we are reexamining the currently available data (see replies #12-14 below). For the other genovariants, we have not identified similar issues. For example, regarding *Ca. Allocryptoplasma*, the genotype (#16) we discovered in *Amblyomma* ticks is novel. Until now, all known *Ca. Allocryptoplasma* strains have been typed (at least) using their 16S rRNA gene sequences, and none exhibit 100% identity to the one identified in our current study.

Was not the genovariant Ehrlichia #6 similar to Candidatus Ehrlichia dumleri? Please, perform another nBLAST search to double check that, since Ca. Ehrlichia dumleri was closely related to an Ehrlichia genotype detect in E. barbara from Brazil.

> Reply #12: Another excellent point, thank you. The updates we made were informative. Here, we conducted additional analyses, similar to those carried out for reply #11, which included novel BLAST searches (see reply #21). We can confirm that *Ehrlichia* genovariant #6 (found in armadillos and tayra of French Guiana) is genetically distinct from *Ca. Ehrlichia dumleri* and the *Ehrlichia* genotype detected in Brazilian tayra. Therefore, they represent different genovariants. However, it's worth noting that *Ehrlichia* genovariant #6 is genetically close to the *Ehrlichia* genotype detected in Brazilian tayra: (1) Based on their 16S rRNA gene sequences, they exhibit 99.74% pairwise nucleotide identities (cf. Supplementary Table 2), (2) Based on their *gltA* gene sequences, they share 99.65% nucleotide identity (GenBank OM055650). This indicates that distinct yet closely related genovariants of *Ehrlichia* are circulating in tayra populations in French Guiana and Brazil. As suggested by the reviewer, *Ehrlichia* genovariant #6 is phylogenetically related to *Ca. Ehrlichia dumleri* (as already

evidenced by the 16S phylogenetic tree presented in Figure 3), as also shown by the *gltA* gene sequences (99.40% nucleotide identity, GenBank OP819940). We have incorporated this information into the results section of the manuscript (lines 205-215). Additionally, we have submitted the *gltA* sequence to GenBank (as mentioned in lines 526-527). We also cite an additional reference, Perles *et al.* (2022; doi: 10.3390/microorganisms10122379), as it first describes *Ca. Ehrlichia dumleri* in Brazil.

What about the genovariant detected in capybaras? Was not it similar to the Candidatus Ehrlichia capybara previously detected in Brazil? Please, check it out: Novel Anaplasmataceae agents Candidatus Ehrlichia hydrochoerus and Anaplasma spp. Infecting Capybaras, Brazil

> Reply #13: We conducted additional analyses to verify this point. Although we did not generate additional data, the results are conclusive based on the 16S rRNA gene sequences: *Ehrlichia* genovariant #5 that we identified in both capybara and *Amblyomma romitii* (a tick species exclusively feeding on capybara) is genetically distinct from *Ca. Ehrlichia hydrochoerus*, as identified in Brazilian capybaras by Vieira *et al.* (2022). According to their 16S rRNA gene sequences, they share only 96.11% pairwise nucleotide identity (over 566 bp; GenBank MW785879 and MW785880 for *Ca. Ehrlichia hydrochoerus*), which is low for two species within the same genus. These *Ehrlichia* genovariants are obviously not the same. We have included this information in the results section of the manuscript (lines 228-233). Additionally, the study referenced by the reviewer, Vieira *et al.* (2022), is now cited in the manuscript.

- Line 179: Did not the authors find high identity to Ehrlichia minasensis in R. microplus ticks?

> Reply #14: The *Ehrlichia* genovariant #9 we found in the *R. microplus* ticks shows a higher pairwise nucleotide identity (99.91%) with *Ehrlichia* sp. VKAA024 (also detected in *R. microplus* ticks, albeit from Malaysia, see Supplementary Table S2) than with *Ehrlichia minasensis* isolate JZT47 and strain B11 (99.90%, GenBank OQ136684 and QOHL01000000, respectively). Nevertheless, we acknowledge that this difference (less than 0.01%) is marginal, and *Ehrlichia* genovariant #9 we found in the *R. microplus* ticks may represent a novel strain of *Ehrlichia minasensis*, albeit distinct from known strains and isolates. Therefore, we have included this information in the results section of the manuscript (lines 189-193).

- Line 199: coelebs

> Reply #15: Typo corrected (line 239).

- Lines 304-305: Regarding the sentence: “The ability of *Ca. Anaplasma sparouinense* to cause intraerythrocytic anaplasmosis in humans shows that there are spillovers from wildlife in Amazon rainforests.” Authors cannot state that, since this genovariant has not been detected in wild animals so far. Please, rephrase or delete this sentence.

> **Reply #16:** Yes, that's accurate. We have revised the sentence in accordance with this suggestion and clarified that the natural host of *Ca. Anaplasma sparouinense* has not yet been identified (lines 354-356).

- Line 311: Please, remove “Amazonian”, since not all the articles cited in this sentence referred to animals sampled in Amazon only.

> **Reply #17:** Done (line 364).

- Line 315: What about the genovariants detected in capybaras? Did they differ from those detected in capybaras from Brazil and Argentina?

> **Reply #18:** Please cf. reply #13.

- Line 319: Be careful when generalizing new genovariants in Brazil as all belonging to Amazon Rainforest! E.g. *Candidatus Anaplasma brasiliensis* was detected in Cerrado biome but not in the Amazon biome. I strongly recommend authors check the biomes where these novel genotypes were described in Brazil.

> **Reply #19:** This is entirely accurate, and we apologize for oversimplifying the previous version of this sentence. Consequently, we have made slight modifications to the sentence to acknowledge the presence of these two biomes in Brazil and to clarify the fact that novel Anaplasmataceae have also been described in the Cerrado biome. The revised sentences now read as follows, which is more precise (lines 370-375):

‘The diversity and endemicity of *Ehrlichia* and *Anaplasma* genovariants in French Guiana, as well as in the northern rainforests of Brazil, suggest that the health hazard is unique in the Amazon biome. The recent detection of additional genovariants in the Cerrado biome, a vast tropical savanna ecoregion of Brazil, further underscores the distinctiveness of South America in the context of tick-borne diseases.’

We have included three additional references in these sentences: Braga *et al.* (2023), André *et al.* (2022), and Vieira *et al.* (2022), all of which were suggested by the reviewer (see replies #11 and #13).

- Line 689: shows

> **Reply #20:** Typo corrected (line 766).

- Supplementary Table 2: Please, include the size (bp) of the sequences obtained for each genovariant, as well as query cover and E-value. Please, indicate the date when nBLAST search was performed.

> **Reply #21:** Excellent suggestion. All these points are now outlined in Table S2.

The original BLASTn search was conducted in August 2023. However, we have since rerun the analysis to ensure greater accuracy. The best match in public databases has changed for *Ehrlichia* genovariants #1 and #10 as shown in Supplementary Table 2 with track changes mode. *Ehrlichia* genovariant #10 now appears as 100% identical to an *Ehrlichia* sp. of Malaysia (Supplementary Table 2, and lines 210-212 in the Results section).

Furthermore, for *Anaplasma* genovariants #13, #14, and #15, we previously listed in Table S2 the best matches based on short sequences available on GenBank. However, BLASTn tends to favor similarities with longer sequences, resulting in different sequences being identified as the best match, despite lower pairwise nucleotide identities. Therefore, for clarity, we have included in Supplementary Table 2 only the best BLASTn matches but we further discuss in the Results section their similarity to other (shorter) sequences (see lines 182-191).

Importantly, this update does not alter the conclusions drawn from Supplementary Table 2: Few of the genovariants we found in French Guiana is 100% identical to genovariants already referenced in GenBank on the basis of their 16S rRNA sequences (see also replies #11-14 for this key point).

- Table 3: What N50 and L50 stand for?

> **Reply #22:** We believe that the reviewer is referring to Table S3, as there is no Table 3. In genomics, N50 and L50 are metrics used to assess the quality and contiguity of sequence assemblies.

N50 is a measure of contiguity and represents the length at which half of the assembled bases are contained in contigs of equal or greater length. It is calculated by arranging all contigs in descending order of length and then determining the length at which the sum of contig lengths equals half of the total genome size. A higher N50 value indicates better assembly quality and longer contiguous sequences.

L50 is the number of contigs required to reach or exceed the N50 value. In other words, L50 represents the minimum number of contigs needed to cover half of the genome. A lower L50 value indicates better assembly contiguity, as it means fewer contigs are needed to reach half of the genome size.

Together, N50 and L50 provide insights into the continuity and completeness of an assembled genome or sequence. For clarity, we now provide a concise description of these significant metrics in the legend of Table S3 (Supplementary Information, lines 15-17).

REVIEWERS' COMMENTS

Reviewer #1 (Remarks to the Author):

Thank you for reworking figures 2 and 3; retaining the full figure in the supplementary is a good compromise.

Again, thank you for making Sup Figure 4. Comments/corrections:

Why are all the genes pointing in the reverse direction in the legend?

ndk is present upstream of tr1 in Florida.

What you have called OpAG4 is actually called OMP1. The OpAGs are part of the msp2 operon, whilst OMP1 is most likely transcribed independently/separately. OMP1 is a member of the pfam01617 family with msp2.

It's a little difficult to distinguish the black from blue in this figure... can you use a lighter blue?

As I said at the beginning, I think this is an interesting

manuscript and provides worthwhile data to an understudied field.

Reviewer #3 (Remarks to the Author):

Authors answered each of the comments raised by this reviewer extraordinarily.

Congrats for such an interesting work!